**Enhancement of the North Atlantic CO₂ sink by Arctic Waters**
Jon Olafsson[1], Solveig R. Olafsdottir[2], Taro Takahashi[3,5], Magnus Danielsen[2] and Thorarinn
S. Arnarson[4,5]
[1] Institute of Earth Sciences, Sturlugata 7 Askja, University of Iceland, IS 101 Reykjavik,
Iceland. jo@hi.is
[2] Marine and Freshwater Research Institute, Fornubúðir 5, IS 220 Hafnafjörður, Iceland
[3] Lamont-Doherty Earth Observatory of Columbia University, Palisades, NY 10964, U.S.A.
[4] National Energy Authority, Grensásvegur 9, IS 108 Reykjavík, Iceland
[5] Deceased
**Abstract**
The North Atlantic north of 50°N is one of the most intense ocean sink areas for atmospheric
$CO_2$ considering the flux per unit area, 0.27 Pg-C yr$^{-1}$, equivalent to -2.5 mol C m$^{-2}$ yr$^{-1}$. The
Northwest Atlantic Ocean is a region with high anthropogenic carbon inventories.  This is on
account of processes which sustain $CO_2$ air-sea fluxes, in particular strong seasonal winds,
ocean heat loss, deep convective mixing and $CO_2$ drawdown by primary production.  The
region is in the northern limb of the Global Thermohaline Circulation, a path for the long term
deep sea sequestration of carbon dioxide. The surface water masses in the  North Atlantic are
of contrasting origins and character, on the one hand the northward flowing North Atlantic
Drift, a Gulf Stream offspring, on the other hand southward moving cold low salinity Polar
and Arctic Waters with signatures from Arctic freshwater sources.  We have studied by
observations, the $CO_2$ air-sea flux of the relevant water masses in the vicinity of Iceland in all
seasons and in different years.  Here we show that the highest ocean $CO_2$ influx is to the
Arctic and Polar waters, respectively, -3.8±0.4 mol C m$^{-2}$ yr$^{-1}$ and -4.4±0.3 mol C m$^{-2}$ yr$^{-1}$.
These waters are $CO_2$ undersaturated in all seasons.  The Atlantic Water is a weak or neutral
sink, near $CO_2$ saturation, after poleward drift from subtropical latitudes. These characteristics
of the three water masses are confirmed by data from observations covering 30 years. We
relate the Polar and Arctic Water persistent undersaturation and $CO_2$ influx to the excess
alkalinity derived from Arctic sources. Carbonate chemistry equilibrium calculations indicate
clearly that the excess alkalinity may support at least 0.058 Pg-C yr$^{-1}$, a significant portion of
the North Atlantic $CO_2$ sink. The Arctic contribution to the North Atlantic $CO_2$ sink which we
reveal is previously unrecognized. However, we point out that there are gaps and conflicts in
the knowledge about the Arctic alkalinity and carbonate budgets and that future trends in the
North Atlantic $CO_2$ sink are connected to developments in the rapidly warming and changing
Arctic. The results we present need to be taken into consideration for the question: Will the
North Atlantic continue to absorb $CO_2$ in the future as it has in the past?

**1 Introduction**
The oceans take up about a quarter of the annual anthropogenic $CO_2$ emissions (Friedlingstein
et al., 2019).  This may even be an underestimate (Watson et al., 2020). The North Atlantic
north of 50°N is one of the most intense ocean sink areas for atmospheric $CO_2$ considering the
flux per unit area (Takahashi et al., 2009). The reasons are strong winds and large natural
partial pressure differences, $\Delta pCO_2 = (pCO_{2sw} - pCO_{2a})$, between the atmosphere and the
surface ocean. The $\Delta pCO_2$ in seawater is a measure of the escaping tendency of $CO_2$ from
seawater to the overlying air.  The $\Delta pCO_2$ is proportional to the concentration of
undissociated $CO_2$ molecules, $[CO_2]aq$, which constitutes about 1 % of the total $CO_2$
dissolved in seawater (the remainders being about 90-95 % as $[HCO_3^-]$ and  4-9 % as $[CO_3^{2-}]$).
The seawater $pCO_2$ depends sensitively on temperature and the $TCO_2$/Alk ratio, the relative
concentrations of total $CO_2$ species dissolved in seawater ($TCO_2 = [CO_2]aq + [HCO_3^-] +$
$[CO_3^{2-}]$) and the alkalinity, Alk,  which reflects the ionic balance in seawater.  Large $\Delta pCO_2$
has been attributed to, a) a cooling effect on the $CO_2$ solubility in the poleward flowing
Atlantic Water, b) an efficient biological drawdown of $pCO_2$ in nutrient rich subpolar waters
and c) high wind speeds over these low $pCO_2$ waters (Takahashi et al., 2002).  Evaluations of
$\Delta pCO_2$ based on observation and models have indicated that the Atlantic north of 50°N and
northward into the Arctic takes up as much as 0.27 Pg-C $yr^{-1}$, equivalent to -2.5 mol C $m^{-2}$ $yr^{-1}$
(Takahashi et al., 2009;Schuster et al., 2013;Landschützer et al., 2013;Mikaloff Fletcher et al.,
2006).  The North Atlantic is a relatively well observed region of the ocean (Takahashi et al.,
2009;Bakker et al., 2016;Reverdin et al., 2018). Nevertheless, estimates of long term trends
for the North Atlantic $CO_2$ sink due to changes in either $\Delta pCO_2$ or wind strength are
conflicting, particularly the Atlantic Water dominated regions (Schuster et al.,
2013;Landschützer et al., 2013;Wanninkhof et al., 2013). The drivers of seasonal flux
variations are considered inadequately understood (Schuster et al., 2013) and a mechanistic
understanding of high latitude $CO_2$ sinks is regarded incomplete (McKinley et al., 2017).  It
has been common to many large scale flux evaluations, modelled or from observations, that
they are based on regions defined by geographical borders, latitude and longitude, e.g.
between 49°N and 76°N for the high latitude Sub Polar North Atlantic (Takahashi et al.,
2009;Schuster et al., 2013).  A more realistic approach is to define biogeographical regions,
biomes (Fay and McKinley, 2014). The influence of oceanographic property differences
within this region on $CO_2$ fluxes has generally not been apparent, primarily due to Arctic
latitude data limitations. The ability of current generation Earth System Models to predict
trends in North Atlantic $CO_2$ has recently been questioned and suggested that their
inadequacies may be caused by biased alkalinity in the simulated background biogeochemical
state (Lebehot et al., 2019).

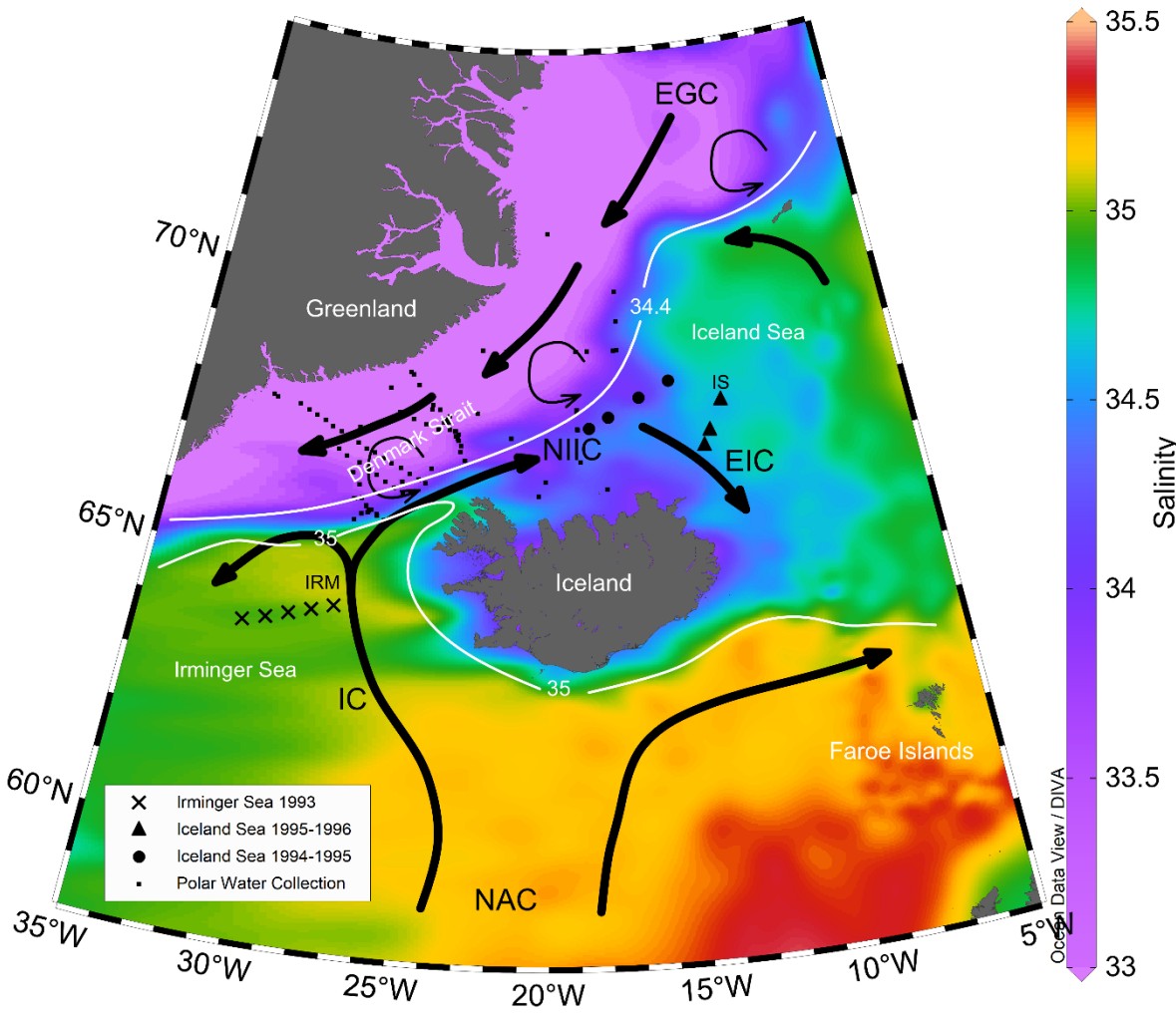


*Figure 1. Mean July to September surface salinity in the vicinity of Iceland.* *The S=35*
*isohaline marks the boundary between northward flowing Atlantic Water and southward*
*flowing cold Arctic Water and low salinity Polar Water. Stations in Irminger Sea marked X*
*and stations in Iceland Sea marked ●for 1994-1995 and ▲for 1995-1996 observations.*
*Collection of Polar Water stations 1983-2012 marked ▪. IRM and IS mark the location of time*
*series stations. NAC: North Atlantic Current, IC: Irminger Current, NIIC: North Iceland*
*Irminger Current, EIC: East Icelandic Current, EGC: East Greenland Current. Map based*
*on the NISE dataset (Nilsen et al., 2008) and drawn using the Ocean Data View program*
*(Schlitzer, 2018).*

The high latitude North Atlantic Ocean in the vicinity of Iceland, is a region of contrasting
surface properties (Fig. 1). The northward flowing North Atlantic Current carries relatively
warm and saline Atlantic Water, derived from the Gulf Stream, as far as the Nordic Seas and
the Arctic Ocean north of Svalbard. The Irminger Current branch carries Atlantic Water to
south and west Iceland and a small branch, the North Icelandic Irminger Current that
transports 1 Sv (1 Sv=$10^6$ m$^3$ s$^{-1}$), reaches the Iceland Sea (Stefánsson, 1962;Våge et al.,
2011). The temperature and salinity properties of the Atlantic Water are known to change
with atmospheric forcing and with freshening events (Dickson et al., 1988;Hátún et al.,
2005;Holliday et al., 2020). The rapid East Greenland Current (EGC) (Håvik et al., 2017)
flows southward from the Arctic to the North Atlantic, carrying Polar Water cold and with
low salinity, S<34.4, due to ice melt and a portion of the large freshwater input to the Arctic
from rivers that contribute about 11% of the global riverine discharge (Sutherland et al.,
2009;McClelland et al., 2012). In between these extremes there are large areas of the
Greenland and Iceland Seas that contain predominantly the intermediate, Arctic Water which
is a product of heat loss and freshwater export from the EGC (Fig. 1) (Våge et al., 2015). The
north- and southward flowing currents are separated by the Arctic Front outlined in Figure 1
by the salinity=35 contour generally oriented SW-NE. Deep water formation in the high
latitude North Atlantic produces cold dense waters which, together with a similar product in
the Labrador Sea, are source waters for the Global Thermohaline Circulation linking the
regional air-sea $CO_2$ flux to a route for the long term deep ocean sequestration of
anthropogenic $CO_2$ (Broecker, 1991). Downstream from the Polar Water and Arctic Water
southward flows is the subpolar North Atlantic with high water column inventories of
anthropogenic carbon (Khatiwala et al., 2013;Gruber et al., 2019). The high anthropogenic
$CO_2$ regions have been attributed to the combined effects of the solubility and biology gas
exchange pumps on the $CO_2$ fluxes (Takahashi et al., 2002). The region of our study
affects large scale ocean-atmosphere $CO_2$ exchange processes in the North Atlantic.
Here we evaluate regional, seasonal and interannual air-sea carbon dioxide fluxes for the main
surface waters characteristic of this region (Fig. 1). We base this work on extensive
observations which cover regional water masses, all seasons and include different states of the
North Atlantic Ocillation, NAO (Flatau et al., 2003). We employ two different observation
approaches for flux estimates. Firstly, repeat station hydrography with emphasis on the
seasonal flux patterns in Atlantic Water and in Arctic Water (Fig. 1).  Secondly, underway
ship records of surface $p$CO$_2$ where the emphasis was on the different surface water masses
(Fig. 2).

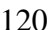


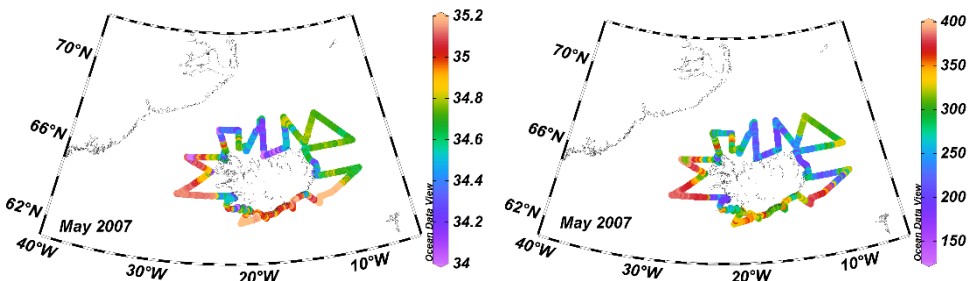

**Figure 2. Cruise tracks where surface layer salinity and $p$CO₂ were recorded**
**underway.** Left sea surface salinity, right $p$CO₂sw(μatm) along the cruise tracks. Maps
drawn using the Ocean Data View program (Schlitzer, 2018).

We describe long term carbon chemistry characteristics of water masses in mid winter when
physical forces prevail over biological processes. For the Irminger Sea and Iceland Sea from
time series observations (Olafsson et al., 2010) and for the EGC Polar Water from a collection
of $p$CO₂ data assembled in the period 1983 to 2012.

**2  Methods**
**2.1  Data acquisition**
**2.1.1  Seasonal studies 1993-1996**
Seasonal carbon chemistry variations in the relatively warm and saline (S>35) Atlantic Water
were studied 1993-1994 on 15 cruises from February 1993 to January 1994 with 5 stations on
a 167 km long transect over the core of the Irminger Current and into the northern Irminger
Sea (Fig. 1 and Tables S1 and S2). In order to close the full annual cycle, to 23 February
1994 we use data from the previous year and date. In 1994-1996 the study centered on the
colder and less saline Arctic Water of the Iceland Sea and was conducted on 22 cruises with
sampling dates from 11 Feb 1994 to 12 Feb 1996


. In 1994 on 4 stations on a 168 km long transect into the Iceland Sea Gyre and in 1995 on 3
stations across the East Icelandic Current (Fig. 1 and Tables S1 and S3 in the supplement).
On each cruise the station work was completed in 1-2 days. For both regions, the timing of
cruises was with the period of the phytoplankton spring bloom in mind (Takahashi et al.,
1993). The work was conducted on vessels operated by the Marine Research Institute (MRI)
in Reykjavik, Iceland, R/V Bjarni Seamundsson and R/V Arni Fridriksson. Three times in
1994 a fishing vessel M/V Solrun, was hired. In August 1994 the stations were completed on
the Norwegian vessel R/V Johann Hjort.
Discrete surface layer, 1m, 5m and 10m, $pCO_2$ samples were collected into 500 ml volumetric
flasks and total dissolved inorganic carbon samples, TCO2, into 250 ml flasks from water
bottles on a Rosette and Sea Bird 911 CTD instruments. The $pCO_2$ samples were preserved
with mercuric chloride and analysed ashore by equilibration at 4°C with a gas of known $CO_2$
concentration followed by gas chromatography with a flame ionization detector. The
instrument was calibrated with $N_2$ reference gas and 3 standards, 197.85 ppm, 362.6 ppm and
811.08 ppm, calibrated against standards certified by NOAA-CMDL at Boulder, CO, USA.
The standards used for the underway measurements were similarly calibrated (Chipman et al.,
1993). Samples for total dissolved inorganic carbon, TCO2, were similarly preserved with
mercuric chloride and analysed by coulometry ashore. Quality assurance and sample storage
experiments indicated an overall precision of the discrete sample $pCO_2$ determinations better
than ±2 µatm and of the TCO2 determinations ±2 µmol kg$^{-1}$ after1990 but ±4 µmol kg$^{-1}$
earlier (Olafsson et al., 2010).

**2.1.2 Underway $pCO_2$ records 2006-2007**
The underway $pCO_2$ determinations in 2006-2007 covered areas of the East Greenland
Current in and northwards from the Denmark Strait, in addition to Atlantic and Arctic Waters.
The 6 cruises (Table S4) covered all seasons and all three water masses but with variable areal
extensions (Fig. 2). Seawater was pumped continuously from an intake at 5 m depth at 10 L
min$^{-1}$ into a shower-head equilibrator with a total volume of 30 L and a headspace of 15 L.
Temperature at the inlet and salinity were measured with an SeaBird Model SBE-21
thermosalinograph (Sea-Bird Electronics, Seattle,WA, USA). Underway $pCO_2$ determinations
were carried out with a system similar to the one described by Bates and coworkers (Bates et
al., 1998). The mole fraction of $CO_2$ (V $CO_2$) in the headspace was determined with a Li-Cor
infrared analyzer Model 6251 (Li-Cor Biosciences, Lincoln, NB, USA). The instrument was
calibrated against four standards of $CO_2$ in air certified by NOAA-CMDL at Boulder, CO,
USA. and a $N_2$ reference gas. The standards had $CO_2$ dry air mole fractions of 122.19,
253.76, 358.41 and 476.81 ppm. The $pCO_2$ sw determinations were corrected to in-situ
seawater temperatures using the equation (Takahashi et al., 1993):
$pCO_2$ sw(in situ) = $pCO_2$ sw(eq) $e^{0.0423(T\text{in situ}-T\text{eq})}$ (eq.1)
The precision of the underway $pCO_2$ determinations is estimated by SOCAT better than ±5
µatm (Bakker et al., 2016).

### 2.1.3 Time series data

We use discrete sample $pCO_2$ andTCO2 data to calculate Total Alkalinity from the Irminger Sea and the Iceland Sea time series stations (Ólafsson, 2012, 2016).

### 2.1.4 Polar Water data collection

Discrete samples for carbon chemistry studies were taken on stations (N=146) in the East Greenland Current when opportunites permitted on cruises in the period 1983 to 2012. The 25 m surface layer data include >400 TCO2 samples and >300 pairs of $pCO_2$ and TCO2 for calculation of carbonate system parameters. The sesonal cycle by month is evaluated from the composite data.

### 2.1.5 Carbonate chemistry calculations

The most desireable way for computing carbonate chemistry parameters is to use $pCO_2$ and TCO2 (Takahashi et al., 2014). We calculate Total alkalinity from discrete sample $pCO_2$ and TCO2 data pairs using the CO2SYS.xls v2.1 software (Lewis and Wallace, 1998;Pierrot et al., 2006) and select carbonic acid dissociation constants (Lueker et al., 2000), the constant for $HSO_4^-$ (Dickson, 1990) and boron concentrations (Lee et al., 2010).

### 2.2 $CO_2$ air-sea flux calculations

In this study, the partial pressure of carbon dioxide in seawater samples has been measured by gas-seawater equilibration methods (Olafsson et al., 2010). The results are expressed as $pCO_2$. The bulk flux of the carbon dioxide across the air-sea interface is often estimated from its relationship with wind speed and sea-air partial pressure difference, $\Delta pCO_2$. We determine the flux (F) from $\Delta pCO_2$ and use Eq. 2 and Eq. 3 for estimating the bulk air-sea fluxes of $CO_2$ (Takahashi et al., 2009)

$$F = k \cdot \alpha \cdot \Delta pCO_2 \qquad \text{(Eq 2)}$$

$$F = 0.251 \, U^2 \, (Sc/660)^{-0.5} \, \alpha \, (pCO_{2 \, w} - pCO_{2 \, a}) \qquad \text{(Eq 3)}$$

There $k = 0.251 \, U^2 \, (Sc/660)^{-0.5}$ is the gas transfer velocity or kinetic component of the expression (Wanninkhof, 2014), $\alpha$ is the solubility of $CO_2$ gas in sea water (Weiss, 1974) and $\Delta pCO_2 = (pCO_{2sw} - pCO_{2a})$, is the partial pressure difference or thermodynamic component of the expression (Takahashi et al., 2009). For the wind speed, U, we use the CCMP-2

reanalysis wind product (Wanninkhof, 2014;Atlas et al., 2011;Wanninkhof and Triñanes,

225     2017).

The atmospheric partial pressure values, $pCO_2{}_a$, used in the $\Delta pCO_2$ calculations are weekly
averages from the GLOBALVIEW-CO2 database for the CO2-ICE location which is at
Vestmannaeyjar islands, off south Iceland (GLOBALVIEW-CO2, 2013). Mauna Loa values
were used for periods where CO2-ICE data was missing, 1983-1992 and 2010-2012 (Tans
and Keeling, 2019). The dry air V $CO_2$ mole fraction values were converted to µatm using
$pCO_2$ (µatm)= V $CO_2$ $(P_a - P_w)$ where $P_a$ is the barometric pressure and $P_w$ is the equilibrium
water vapour pressure (Weiss and Price, 1980).
For the Irminger Sea seasonal study we use 30 day running means of the squared daily wind
speed for the region 63.5°N to 64.5°N and 27°W to 32°W and for the Iceland Sea seasonal
study a similar wind product for the region 66.5°N to 68.5°N and 12°W to 19°W. Fluxes
were calculated for the periods between cruises from interpolated $pCO_2$ data and period mean
30 day squared wind running means data. There are thus 14 flux periods covering a year for
the Irminger sea and 21 flux periods covering two years in the Iceland Sea (Tables S1 and S2
in the supplement). The annual fluxes were found by summation of the period fluxes (Table

240     1).

For the underway cruises 2006 to 2007 we used CCMP-2 daily wind fields at 1x1 degree for
the region 62°N to 72°N and 5°W to 40°W. This region was further divided into 4 sub-
regions by latitude 64.9°N and longitude 20°W. Daily 30 day running means of the squared
wind speed from two locations in each sub-region were extracted and their means used for
flux calculations when the vessel sailed in the area. Fluxes were calculated for all $pCO_2$ data
from the 6 cruises, in total 42938 measurements.
The flux data from each of the 6 cruises were categorized into the three sea water types using
the following criteria:

249        1) Atlantic Water S>35, Arctic Water S: 34.4-34.9, Polar Water S<34.4.

250        2) Seasonal salinity and temperature variations were taken into account.

251        3) Waters with runoff influences from Iceland were excluded using salinity and ship

252           position data.

Thus a total of 33352 measurements were used, or 78% of the flux data points. The $CO_2$
fluxes in the realm of each water mass were assessed for the duration of each cruise by
numerical integration. Fluxes in the 5 periods between cruises were assessed by interpolation
of temperature, salinity and $pCO_2$ for each water mass and by using period regional 30 day
running means of squared wind speed data. The annual flux for each water mass  was assessed
by summation.

**3  Results**
**3.1  Seasonal variations and annual CO₂ fluxes at regional water masses**
The wind gas transfer coefficient reveals seasonal variations reflecting strong winds in winter
when they may be stronger over the Irminger Sea than the Iceland Sea as in 1993-1994 and in
1994-1995 (Fig. 3a, Fig. S1).  Both the Irminger Sea and the Iceland Sea seasonal studies
reveal the stongest $CO_2$ undersaturation, with negative $\Delta pCO_2$ of about 100 µatm in May at
the time of the phytoplankton spring bloom (Fig. 3b). The undersaturation diminishes through
the summer and autumn followed by a gradual return to winter conditions (Takahashi et al.,
1985;Peng et al., 1987;Takahashi et al., 1993).

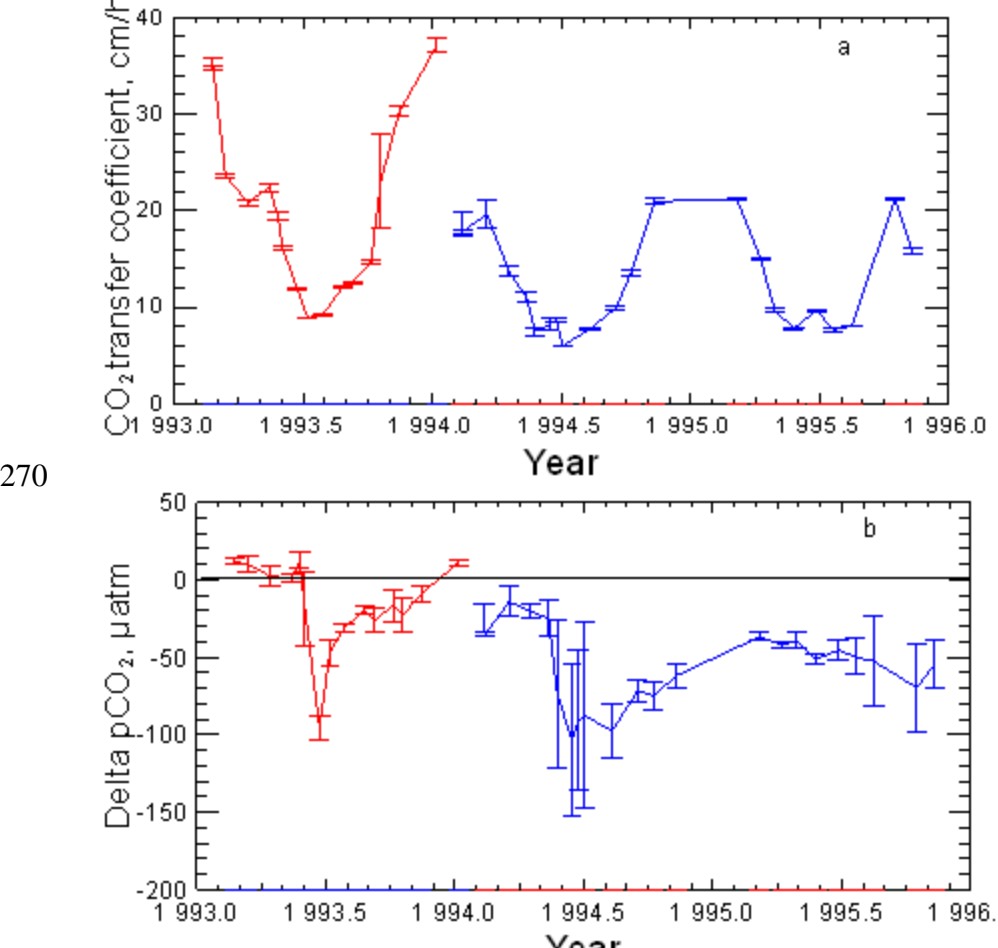

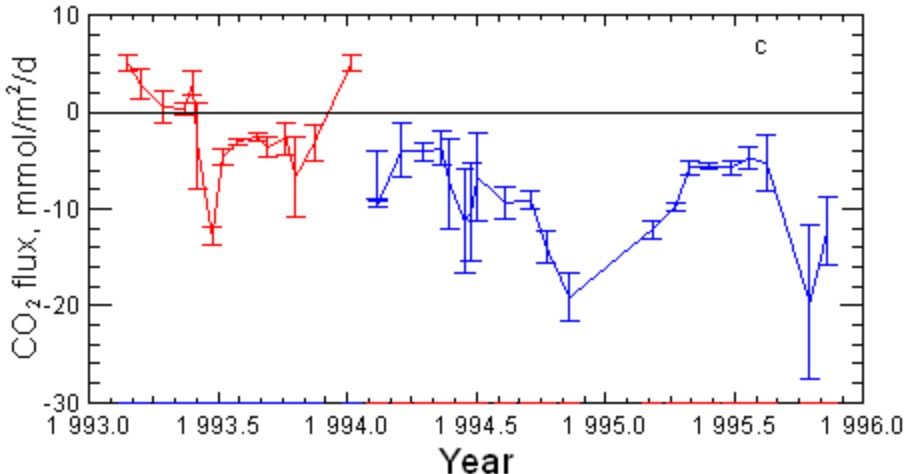

*Figure 3.* *Seasonal variations in the Atlantic Water of the Irminger Sea (red) and in*
*the Arctic Water of the Iceland Sea (blue).* *The gas transfer velocity (a) reflects the seasonal*
*wind strength and the error bars its variations during intervals between cruises. Delta pCO$_2$*
*(b) records the tendency for CO$_2$ to be transferred to the atmosphere (positive) or from the*
*atmosphere to the ocean (negative). The CO$_2$ flux rate (c) reveals that the Arctic Water is a*
*CO$_2$ sink in all seasons whereas the Atlantic Water is a source in winter and a weak sink at*
*other times of the year. The error bars indicate ± 1 standard deviation from the mean and*
*reflect the variations between the stations observed each cruise.*

The CO$_2$ influx in the spring is, however, relatively small as the wind gas transfer coefficient
is then moderate (Fig. 3a). In the autumn the winds strengthen with heat loss and vertical
mixing while CO$_2$ undersaturation still persists. In mid winter, February-March, vertical
mixing brings richer CO$_2$ water to the surface of the Irminger Sea leading to supersaturation
(Ólafsson, 2003), the flux reverses and the region becomes a weak source for atmospheric
CO$_2$ (Fig. 3c).

**Table 1** **Annual sea–air CO$_2$ fluxes (mol C m$^{-2}$ y$^{-1}$) in the three water masses.**

| Water masses and evaluation methods | CO$_2$ flux mol C m$^{-2}$ y$^{-1}$ |
|---|---|
| Atlantic water, repeat stations 1993 | -0.69±0.16 |
| Atlantic water, Underway Measurements, 2006-2007 | 0.07 ± 0.15 |
| Arctic water, repeat stations 1994 | -3.97±0.48 |
| Arctic water, repeat stations 1995 | -3.60±0.31 |
| Arctic water, Underway Measurements, 2006-2007 | -2.84 ± 0.19 |
| Polar water, Underway Measurements, 2006-2007 | -4.44 ± 0.34 |


The integrated annual $CO_2$ flux shows that the Atlantic Water in the Irminger Sea was a weak
sink, $-0.69\pm0.16$ mol C m$^{-2}$ y$^{-1}$, in 1993 (Table 1). The more extensive underway area
coverage of the Atlantic Water in 2006-2007, confirmed in essence the seasonal pattern and
indicated that the Atlantic Water was a neutral sink, $0.07 \pm 0.15$ mol C m$^{-2}$ y$^{-1}$ for this year
(Table 1). The winter gas transfer coefficient was again significantly larger over the Atlantic
Water regions than the Arctic and Polar Waters, facilitating air-sea equilibration (Fig. 4b).
The years of the Iceland Sea observations, 1994-1996, coincided with a large transition in the
North Atlantic Ocillation (NAO) from a positive state 1994/1995  to a negative state in
1995/1996 and large scale shifts in ocean fronts (Flatau et al., 2003).  Vertical density
distribution in the Iceland Sea indicated an enhanced convective activity in 1995 (Våge et al.,
2015).  Cold northeasterly winds were persistent in the spring of 1995 resulting in record low
temperature anomalies for the north Iceland shelf (Ólafsson, 1999).  In 1995 the spring bloom
associated undersaturation, $\Delta pCO_2$,  was only half of that in 1994, possibly due to a weaker
stratification in May and continued over the summer season (Fig.S2) (Våge et al., 2015).  As
in the Irminger Sea the spring bloom associated $CO_2$ influx is small. The largest $CO_2$ influx
was in the fall and early winters of 1995 and 1996 as temperature dropped, winds gathered
strength and vertical mixing was enhanced.  This compensated for the small spring bloom in
1995 and the annual bulk fluxes 1994 and 1995 are similar and high despite very different
physical conditions (Table 1).  The UW$p$CO$_2$ surveys had less temporal resolution but
confirmed all year undersaturation of the Arctic Water. However, the integrated annual influx,
$-2.84$ mol C m$^{-2}$ y$^{-1}$, was significantly less than evaluated with repeat station data even though
the strength of the gas transfer coefficient was similar in both studies (Table 1, Figs.4a and
4b). This may reflect the large underway area coverage compared with the repeated fixed
stations.

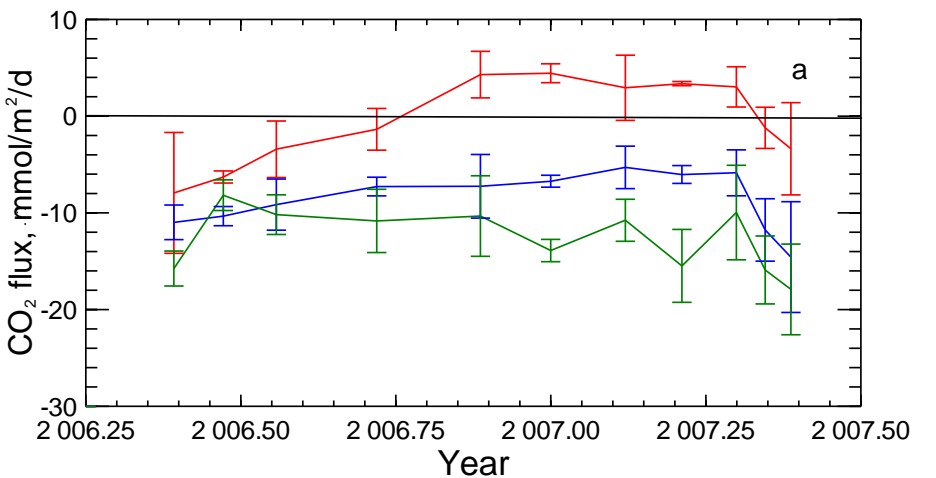


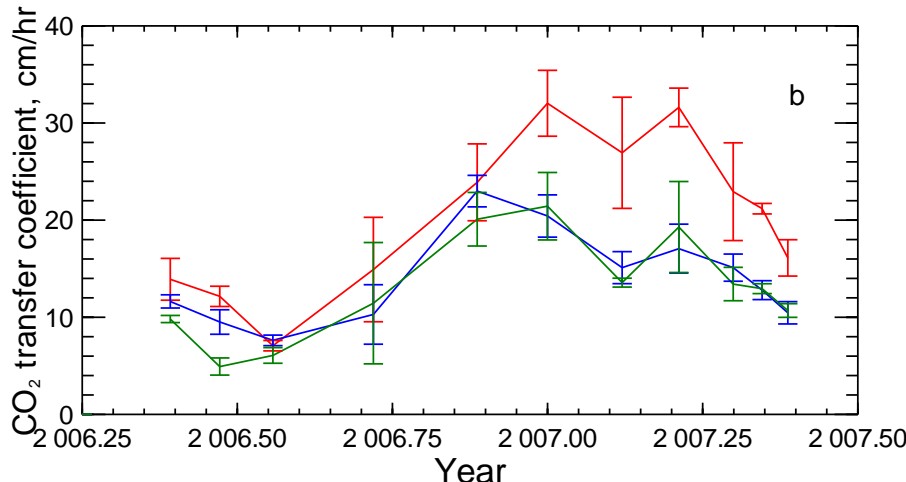


**Figure 4. Seasonal air-sea CO₂ flux variations from UWpCO₂ observations**.
*a)Atlantic Water (red) is a weak sink in summer and neutral over the year, n=7068. Both*
*Arctic Water (blue) n=16874, and Polar Water (green) n=9410, are strong sinks throughout*
*the year. The error bars indicate ± 1 standard deviation from the mean. b) The gas transfer*
*coefficient for the Atlantic Water regime is significantly stronger in winter than for the Arctic*
*and Polar Water.*

Ice cover in the East Greenland Current is variable and the ice edge at the seasonal minimum

has moved northward and from the Denmark Strait with decreasing Arctic sea ice (Serreze

and Meier, 2019). The Polar Water salinity ranges from 34.4 to less than 30 in summer. The

lowest salinity water freezes leading to salinity around 34 in winter. We covered the Polar

Water in all six UW$p$CO₂ surveys 2006-2007 (Fig. 2) and undersaturation characterised this

water mass in all cruises. The integrated annual influx, -4.44 mol C m$^{-2}$ y$^{-1}$ (Table 1, Fig.4),

shows the Polar Water to be the strongest CO₂ sink, 80 % above the estimated mean for the

Atlantic north of 50°N, -2.5 mol C m$^{-2}$ y$^{-}$(Takahashi et al., 2009). Further comparison with

the Takahashi climatology indicates a broad agreement with Arctic Water region NE of

Iceland with -3.5 to -4.5 mol C m$^{-2}$ y$^{-1}$ and with the Atlantic Water region S and SW of

Iceland with about -1 mol C m$^{-2}$ y$^{-1}$ (Takahashi et al., 2009).

**3.2 Long term Δ$p$CO₂ characteristics of the regional water masses**

We evaluate the long term $p$CO₂ characteristics of the three water masses from three other

data assembled over about 30 years. We use the Polar Water data collection and draft a

composite picture of seasonal Δ$p$CO₂ variations in Polar Water in and north of the Denmark

Strait (Fig.1) which confirms all year undersaturation, deep in summer, and in mid winter

when salinity raises to ~ 34, the Δ$p$CO₂ levels at about -50 µatm (Fig. 5a). Long term winter

Δ$p$CO₂ in the Irminger Sea and Iceland Sea (Figs. 1 and 5b) when biological activity is

minimal (Olafsson et al., 2009), show the Atlantic Water to be slightly supersaturated and

following the atmospheric pCO₂ increase of 1.80 µatm/yr, whereas the Arctic Water is
undersaturated to about -35 µatm. The Gulf Stream derived Atlantic Water which reaches the
northern Irminger Sea and the Nordic Seas, has had a long contact time with the atmosphere
to loose heat and reach near $CO_2$ saturation (Takahashi et al., 2002;Olsen et al., 2006).

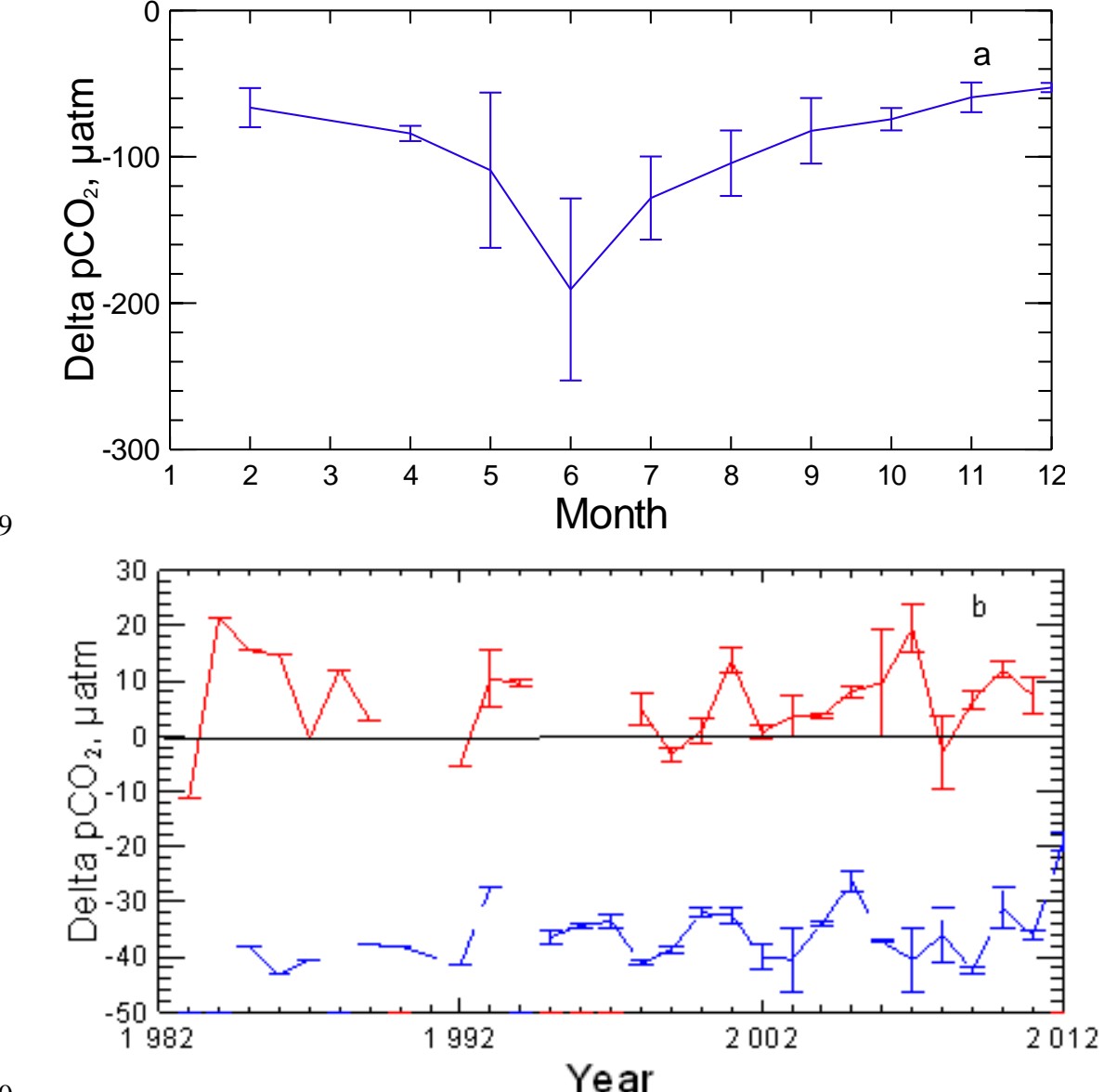



***Figure 5. Water mass decadal surface water pCO₂ characteristics.*** *a) A composite picture of*
*Delta pCO₂ from 146 stations with Polar Water pCO₂ observations (n=312) from the 25 m*
*surface layer 1983 to 2012 shows undersaturation at all times of the year. The error bars*
*indicate ± 1 standard deviation from the monthly means. b) Atlantic Water at the Irminger*
*Sea time series station (red) is generally a weak CO₂ source in winter (24 winters, 52*
*samples), January-March, whereas winter (25 winters, 61 samples) CO₂ undersaturation*
*persits at the Iceland Sea time series site (blue). The error bars indicate ± 1 standard*
*deviation from the surface layer station means.*

The Polar Water in the East Greenland Current which is advected southward from the Arctic
is in general characterised by low temperature and large seasonal salinity and carbonate
chemistry variations. Both physical and biogeochemical processes generate the large seasonal
variability but the winter observations represent the state of lowest biological activity (Fig. 6)
(Table 2).  The TCO2 data in Table 2 are uncorrected for hydrographic variations or
anthropogenic trends but the Atlantic Water is based on a short period of 10 years and the
Polar Water atmospheric contact history is poorly known.

**Table 2.  Mean IRM-TS Atlantic Water surface layer conditions in winter, 2001-2010**
**and in Polar Water 25 m surface layer November to April 1984-2012.**

| | T °C | Salinity, S | Density, $\rho$ kg m$^{-3}$ | TCO2/S $\mu$mol kg$^{-1}$ psu$^{-1}$ | ALK/S $\mu$mol kg$^{-1}$ psu$^{-1}$ | TCO2/ALK | $p$CO$_2$ $\mu$atm |
|---|---|---|---|---|---|---|---|
| Atlantic Water | 7.11 ± 0.36 | 35.13 ± 0.03 | 1027.507 ± 0.034 | 61.11 ± 0.09 | 65.96 ± 0.13 | 0.926 ± 0.002 | 388 ±9 |
| Polar Water | -0.31 ±1.53 | 33.95 ± 0.33 | 1027.255 ± 0.244 | 62.16 ±0.54 | 66.49 ± 0.40 | 0.935 ± 0.004 | 301 ±11 |


The winter conditions in the northward flowing Atlantic Water at the Irminger Sea time series
station 2001-2010 (Table 2) are in stark contrast and with notably higher $p$CO$_2$ and lower
TCO2/S and ALK/ S ratios than the Polar Water in winter.

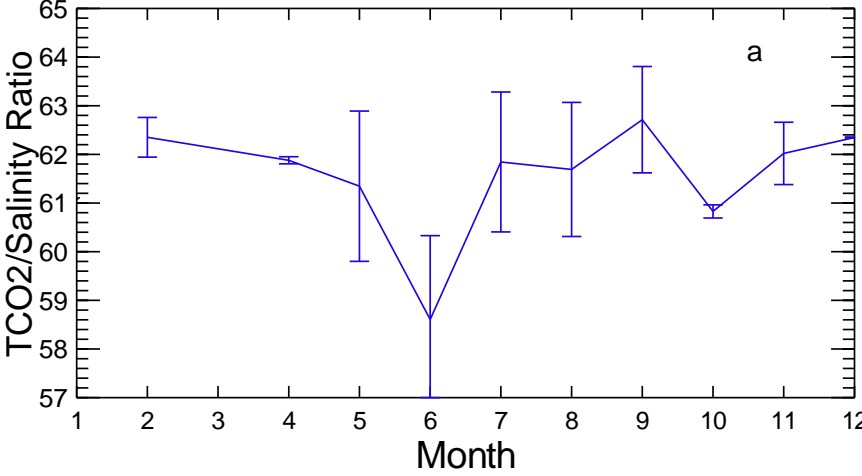


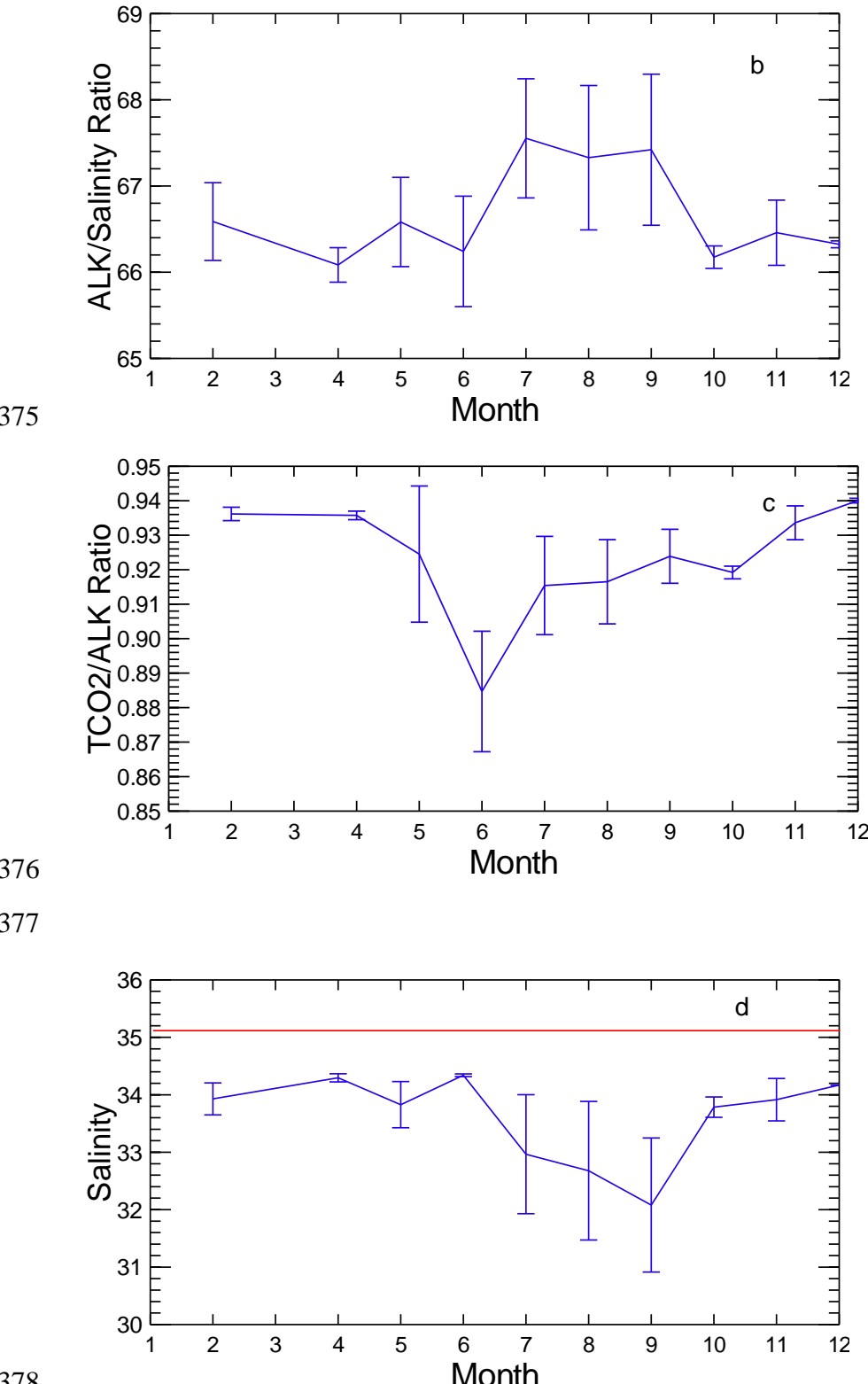





**Figure 6. Polar Water seasonal carbonate chemistry variations.** Composite Polar Water
data from the 25m surface layer. The seasonal variations in the a) Total inorganic
carbon/Salinity ratio, μmol kg$^{-1}$ psu$^{-1}$, b) Alkalinity/Salinity ratio, μmol kg$^{-1}$ psu$^{-1}$, and c)
Total inorganic carbon/Alkalinity ratio reflect biological carbon assimilation and inorganic
processes associated with fresh water inputs which lower the salinity d) to the annual

minimum in late summer.  The red horizontal lines mark the Atlantic Water benchmarks
(Table 2).

We take the Atlantic Water in winter (Table 2) as a proxy (benchmark) for the relatively
warm and saline water advected from the North Atlantic to the Nordic Seas and the Arctic and
compare  with it the carbonate chemistry seasonal variations in the southward flowing Polar
Water (Fig. 6).  The ALK/S ratio for the Polar Water is higher than that for the Atlantic Water
in winter and throughout the year (Fig. 6b).  The TCO2/S ratio of the Polar Water is larger
than that of the Atlantic Water except in early summer when biological assimilation,
photosynthesis, decreases the TCO2 concentration.  The TCO2/ALK ratio falls as a
consequence (Fig. 6c) which leads to strong $p$CO$_2$ undersaturation and large Delta $p$CO$_2$
(Figs 6c and 5a). The high TCO2/S and ALK/S ratios  indicate alkalinity and carbonate inputs
as freshwater lowers the Polar Water salinity to a minimum in late summer (Fig. 6d).

**4  Discussion**
The Polar Water TCO2/S and ALK/S ratios (Table 2 and Fig. 6) indicate both alkalinity and
dissolved carbonate additions.  The choice of winter ratios (Table 2) as benchmarks is solely
for the evaluation of seasonal changes in the Polar Water. Representative annual long term
TCO2/S and ALK/S means would be more realistic but are not available.  Still, such a TCO2/
S ratio would expectedly be lower than the winter one.  An assessment of the effects of the
relative TCO2 and ALK additions to Polar Water depends on the benchmarks chosen (Table

405     2) .

The carbonate chemistry of Polar Water differs from that of open ocean waters, e.g. Atlantic
Water, in having an increasingly higher alkalinity/salinity ratio+ as the salinity decreases from
about S=34.4.  The excess alkalinity has been attributed to the high riverine input from
continents to the Arctic (Anderson et al., 2004). The flow-weighted average alkalinity of 6
major Arctic rivers, discharging 2.245 x 10$^3$ km$^3$ yr$^{-1}$, is 1048 µmol kg$^{-1}$, however, without
assessed uncertainty (Cooper et al., 2008). The river runoff into the Arctic is estimated to be
about 4.2 x 10$^3$ km$^3$ yr$^{-1}$, or 0.133 x 10$^6$ m$^3$ s$^{-1}$ (0.133 Sv).  This is about 11% of the global
freshwater input to the oceans (Carmack et al., 2016).  Taking the average alkalinity 1048
µmol kg$^{-1}$, the amount of alkalinity added by rivers to the Arctic and transported to the North
Atlantic via the Canadian Arctic Archipelago and via the Fram Strait and further south with
theLabrador and East Greenland Currents, would be 4.4 x 10$^{12}$ mol yr$^{-1}$ (Supplement). Cooper
et al. (2008) reported on riverine alkalinity but not on associated inorganic carbonate.  A
recent assessment of Polar Water boron concentrations indicates insignificant borate
contribution with Arctic rivers (Olafsson et al., 2020). The riverine alkalinity may primarity
be attributed to carbonate alkalinity, $CA=[HCO_3^-]+2[CO_3^{2-}]$. The potential of the added
alkalinity to reduce $p$CO$_2$ of seawater would depend on its excess over TCO2.
Linear alkalinity-salinity relationships observed in the Arctic Ocean and the Nordic Seas and
their extrapolated intercepts to S=0, have indicated freshwater sources with alkalinity 1412
µmol kg$^{-1}$ (Anderson et al., 2004) and 1752 µmol kg$^{-1}$ (Nondal et al., 2009). Climatological
data from the West- Greenland, Iceland and Norwegian Seas show a high S=0 intercept of
1796 µmol kg$^{-1}$ but a lower one for the High Arctic north of 80°N, 1341 µmol kg$^{-1}$ (Takahashi
et al., 2014). The climatological relationships were for Potential Alkalinity, PA=TA + NO$_3^-$,
which has little influence since the nitrate concentrations are low. The intercepts may be
interpreted as the mean alkalinity of fresh waters added to the Arctic by rivers and melting ice
and snow. However, the above intercepts indicate considerable variability, they are also
higher than the average alkalinity of Arctic rivers, 1048 µmol kg$^{-1}$. The excess alkalinity
would lower the $p$CO$_2$ in seawater (and increase the pH), and thus give it an increased
capacity to take up CO$_2$ from the air. The thermodynamic driving force for seawater CO$_2$
uptake, ($p$CO$_{2sw}$ – $p$CO$_{2a}$), would be enhanced.
How large is the potential effect of excess Arctic alkalinity on the CO$_2$ uptake by the Nordic
Seas and the North Atlantic? We consider an estimate. Suppose that the $p$CO$_2$ in seawater
was restored to the original value by absorbing CO$_2$ from the atmosphere. The carbonate
equilibrium relations in seawater give that $p$CO$_2$ is unchanged if ΔTCO$_2$ /ΔAlk = 0.85 Fig. S3
and Supplement. This ratio of additions is nearly constant in the temperature and salinity
range of the subarctic North Atlantic surface waters (t=5°C, S=35). The volume transport of
Polar Water, density <1027.8 kg m$^{-3}$, by the EGC has recently been estimated as 3.9 Sv (Våge
et al., 2013). Taking S=33.0 for the mean Polar Water salinity and using Equations 6 and 7 in
Nondal et al (2009), the mean Polar Water alkalinity is 2256 µmol kg$^{-1}$ which is 46 µmol kg$^{-1}$
more alkalinity than for Atlantic Water calculated at the same salinity (Nondal et al., 2009).
This much excess alkalinity would lower the $p$CO$_2$ of Atlantic Water by 88 µatm and
increase the pH by 0.10. Thus, the excess alkalinity advected to the North Atlantic by the
EGC is 5.7 x 10$^{12}$ mol yr$^{-1}$. Using 0.85 for the ΔTCO$_2$/ΔAlk additions at a constant $p$CO$_2$, we
obtain that the contribution of the excess EGC alkalinity to the uptake of CO$_2$ from the
atmosphere would be 4.8 x 10$^{12}$ mol CO$_2$ yr$^{-1}$, or 0.058 Pg-C yr$^{-1}$. The estimate corresponds
to 21 % of the net CO$_2$ uptake of 0.27 Pg-C yr$^{-1}$ for the subarctic oceans north of 50 °N
(Takahashi et al., 2009). We did not include in the estimate any alkalinity contribution with
the considerable Canadian Arctic Archipelago Polar Water transport (Haine et al., 2015). The
effect of excess alkalinity on the North Atlantic $CO_2$ uptake flux may therefore be
substantially greater than our estimate. We note that the winter undersaturation levels, of -50
µatm and -35 µatm observed in the Polar and Arctic Waters, respectively (Fig. 5), translate to
excess alkalinity of 19 µmol $kg^{-1}$ and 21 µmol $kg^{-1}$ for further $CO_2$ influx downstream.
The difference between the average measured Arctic river alkalinity and the regression based
estimates of alkalinity sources suggests that other origins and processes than the rivers
contribute to the Polar Water alkalinity exported with currents from the Arctic to the Atlantic
Ocean. Photic layer primary production in the absence of calcification may lower the
TCO2/Alk ratio and seawater $p$CO$_2$ in marginal seas (Bates, 2006), while acidification is
increasing in other regions (Anderson et al., 2017;Qi et al., 2017) and projected to become
extensive at the end of the century (Terhaar et al., 2020). Furthermore, the sea-ice seasonal
formation and melting may affect the TCO2/Alk ratio (Grimm et al., 2016;Rysgaard et al.,
2007). Efforts to reconstruct alkalinity fields and alkalinity climatology for the Arctic have
however been difficult (Broullón et al., 2019).

The Arctic is complex and complex climate warming related changes are observed in the
western Arctic Ocean (Ouyang et al., 2020) and expected in marine freshwater systems of the
warming Arctic (Carmack et al., 2016). Not least is the ice cover and areas of multi-year ice
decreasing (Serreze and Meier, 2019). River water alkalinity increases with an addition of
cations derived from the chemical weathering of silicate and carbonate rocks (Berner and
Berner, 1987). Accordingly, an increase in Arctic weathering rates, in response to warmer
climate and increasing atmospheric $CO_2$, could increase the river water alkalinity transported
into the oceans. Such an increase would lower the $p$CO$_2$ in seawater and enhance the oceanic
uptake of atmospheric $CO_2$, providing a negative feedback mechanism to the climatic
warming resulting from increased atmospheric $CO_2$.

**5 Conclusions**
The North Atlantic region we describe has Atlantic Waters advected from southern temperate
latitudes and cold lower salinity Arctic and Polar Waters carried with the East Greenland
Current from the Arctic. The Atlantic Water seasonal $p$CO$_2$ variations are primarily driven by
regional thermal and biological cycles but without much net annual influx of $CO_2$. The
southward flowing Arctic and Polar Waters are on the contrary strong and persistent all year
$CO_2$ sinks. These waters are advected towards the sub-polar North Atlantic with high
inventories of anthropogenic carbon. The TCO2/S and ALK/S Polar Water ratios are higher
than those for the Atlantic Water indicating carbonate and alkalinity sources. We point here to
the Polar Water and Arctic Water $CO_2$ influx and excess alkalinity as an additional
unrecognized source contributing to the North Atlantic $CO_2$ sink. We also see that there are
gaps and conflicts in the knowledge about the Arctic alkalinity and carbonate budgets and that
future trends in the North Atlantic $CO_2$ sink are connected to developments in the rapidly
warming and changing Arctic.

**Acknowledgements**
The NMR Nordic Environmental Research Programme: Carbon Cycle and Convection in the
Nordic Seas, supported the Marine Research Institute (MRI), Reykjavik, repeat station study
in 1993-1995. The MRI work in 2006-2008 was supported by the European Union 6th
Framework Program CARBOOCEAN, EU Contract: 511176. Taro Takahashi was supported
to work on the manuscript with a grant from the the US National Oceanographic and
Atmospheric Administration. The CCMP-2 wind product was generously provided from
Remote Sensing Systems (www.remss.com/measurements/CCMP) by Dr. Joaquin Triñanes
of CIMAS/AOML, Miami.  We gratefully acknowledge the long term technical support from
John Goddard and Tim Newberger, Lamont-Doherty Earth Observatory. We are grateful for
the invaluable cooperation we have had with the crews of all vessels operated in this study
and to Norwegian colleagues for providing time for station work in August 1994.


*Author Contributions.*  J.O., T.T. and S.R.O.  wrote the manuscript.  J.O., Th.S.A., S.R.O. and
M.D. conducted the fieldwork.  J.O., T.T. S.R.O. and Th.S.A., conceived this study.

*Competing interests.*  The authors declare no competing financial interests.

*Data availability.*
The underway $p$CO$_2$ data is available at Ocean Carbon Data System (OCADS)  (Takahashi et
al., 2019).  The Irminger Sea and Iceland Sea seasonal study data and the Polar Water
collection data are stored at the Marine and Freshwater Research Institute, Reykjavik and
available by request. Irminger Sea and Iceland Sea time series data for calculation of Delta
$p$CO$_2$ in winter is at NOAA National Centers for Environmental Information (Ólafsson, 2016,

519  2012).

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
