# Peer review of "Enhancement of the North Atlantic CO2 sink by Arctic Waters"

_Biogeosciences, 2020_

## Referee Comment (RC1) · Anonymous Referee #1 · 10 Sep 2020

Review of the manuscript submitted to Biogeosciences, MS No.: bg-2020-313. "Enhancement of the North Atlantic CO2 sink by Arctic Waters" by Olafsson et al.,

Review invitation: 5/9/20 Review accepted: 7/9/20 Review sent: 10/9/20

General comment:

The Ocean is a major CO2 sink, representing about 25% of the total anthropogenic emissions (Le Friedlingstein et al 2019), but how this carbon sink varies at interannual to decadal scales is still subject to large uncertainties. Even the mean global ocean CO2 sink is still subject to uncertainties. In a recent study Watson et al (2020) re-evaluated a global ocean carbon sink that would reach 3.7 PgC/yr in 2018, or 4.2 PgC/yr for the anthropogenic ocean CO2 sink, i.e. 36 % of the total emissions in 2018

(fossil fuel + land used of 11.5 PgC/yr, Friedlingstein et al 2019) a number close to the results of 31 % based on CO2 interior inventories but for the period 1994-2007 (Gruber et al 2019). At the end of the abstract authors ask the question: "Will the North Atlantic continue to absorb CO2 in the future as it has in the past?" Interestingly, Gruber et al (2019) suggested a reduction of the inventory change in the North Atlantic but this is somehow limited to available data. As recalled by Olafsson et al, it is well-known that the high latitudes in the North Atlantic, north of 50°N represent a strong CO2 sink, both in term of air-sea CO2 flux and anthropogenic CO2 inventories (e.g. Peng et al 1987; Takahashi et al 1993, 2002, 2009; Sabine et al 2004; Khatiwala et al 2013). The results presented by Olafsson et al confirm these previous findings, a large seasonality of fluxes (or pCO2) and contrasting ocean carbon sink in North Atlantic Drift, Polar and Arctic surface waters. Here they focus on these waters observed around Iceland using historical (over 30 years) and more recent pCO2 observations and suggest that ALK input from the Arctic explain part of the contrasting CO2 flux estimated in these water masses. The analysis is mainly based on 1994-1995 time-series and 2006-2007 pCO2 underway observations conducted around Iceland, and extended to a synthesis of 30 years of observations in this region (although it is not clear what data are used over 30 years). The paper is organized in two parts. First authors evaluate the seasonal and interannual (over 30 years) air-sea carbon dioxide fluxes in 3 waters masses characteristic of this region. Second they investigate the origin of the differences of the flux and based on ALK budgets suggest that excess alkalinity from Arctic is a significant source contributing to the North Atlantic CO2 sink.

The first part of the analysis (seasonality and CO2 fluxes) is not really new, although I think it is the first time the air-sea CO2 fluxes are calculated with the data obtained in 2006-2007 around Iceland (these data were added this year in SOCAT-v2020). What is new concerns the impact of Alkalinity (ALK) to explain the difference of DpCO2 and CO2 sink/source. It is suggested that excess alkalinity derived from Arctic sources contribute significantly to the CO2 sink in this region. This is an interesting analysis, as coupled climate/carbon models generally failed in reproducing correctly the ALK fields

and this might impact on future CO2 sink estimates especially in the North Atlantic (e.g. Lebehot et al 2019), not only linked to future changes in E/P at large scale but also potential dramatic physical and biogeochemical changes in the Artic in the future (Terhaar et al 2020). The topic addressed in this paper is in line for Biogeosciences. However, the manuscript needs revision, especially regarding ALK observations. Nowhere in the manuscript have the authors presented ALK data that would support the analysis. They mostly refer to previously published ALK/salinity relationships and mean ALK in Arctic rivers. Also, Supplementary Figures are missing in the file (?) and this would probably help to follow the discussion for the main new results presented. In the conclusion, a schematic view of the ALK input in the region (with uncertainty) would be nice and show how this is linked to the CO2 sink around Iceland.

Overall, the study by Olafsson and co-authors by revisiting historical observations and presenting new data (2006-2007) presents interesting results in this region that open new questions regarding the oceanic carbon sink in relation to water masses and alkalinity changes and is suitable for publication after major revision.

Other comments (including minors), suggestions and questions are listed below:

;;;;;;;;;;;;;;;;;;;;; Specific and minor comments

C-01: Page 2, Lines 57-62. Authors write: "Estimates of long term trends for the North Atlantic CO2 sink due to changes in either DpCO2 or wind strength are conflicting, particularly the Atlantic Water dominated regions (Schuster et al., 2013; Landschützer et al., 2013; Wanninkhof et al., 2013). The drivers of seasonal flux variations are inadequately understood (Schuster et al., 2013) and a mechanistic understanding of high latitude CO2 sinks is considered incomplete (McKinley et al., 2017)." Well, I think compared to other regions, the seasonal variations (Peng et al 1987; Takahashi et al 1993, 2002) and long term trends of pCO2 are relatively better observed in the North Atlantic, although it has been recognized that annual trends are also sensitive to interannual variability (including linked or not to NAO or AMO/AMV) and the trends

better detected during winter when productivity is low (e.g. Metzl et al 2010; Frob et al, 2019). This is probably the reason why authors used winter data over 30 years to produce figure 6b.

C-02: Page X, Lines 62-65: Authors write: "It is common to many large scale flux evaluations, modelled or from observations, that they are based on regions defined by geographical borders, latitude and longitude, e.g. between 49°N and 76°N for the high latitude Sub Polar North Atlantic (Takahashi et al., 2009;Schuster et al., 2013)." The choice of latitudinal bands was mainly selected based on the "big boxes" used in atmospheric inversions (TRANSCOM) and compare ocean estimates versus inversions (e.g. specific boxes for RECAPP, Schuster et al 2013). However, for ocean purpose, many products are now based on "biomes" definition (Fay and McKinley, 2014) and used for air-sea CO2 fluxes calculations (and trends) from reconstructed pCO2 fields (e.g. SO-COM project, Rödenbeck et al 2015; Landschützer et al, 2016; Denvil-Sommer et al 2019). The region investigated by Olafsson et al, is at the boundary of biomes NA-ICE and NA-SPSS (see Fay and McKinley, 2014) but results in biome NA-ICE are often omitted due to sparse data coverage (Rödenbeck et al 2015). However, methods are now able to add pCO2 climatology in the Arctic considered as a single "biome" (Landschützer et al, 2020). It might be interesting to compare the seasonal view presented here (Figure 6) with most recent pCO2 climatology around Iceland (maybe for another paper).

C-03, Page 4, Line 84 and Introduction: Concerning the circulation in the high latitudes in the north Atlantic (and freshwater input from the Arctic) I would suggest to refer to Holliday et al (2020) who described the observed recent changes in salinity in this region, that probably also impact on surface waters properties observed around Iceland (including TALK and pCO2 and CO2 sink ?).

C-04: Page 4, Line 106: Would be interesting to show NAO (and AO) index, e.g. for the full 30 years period (1982-2012, to add in figure 6 ?) and discuss if you find a link (or not) with NAO as this is still debatable. This is optional as variability of NAO is not

really discussed in the manuscript (excepted for 1994-1996 period).

C-05: Page 4-5: Figure 2 shows the underway pCO2 data for 2006-2007. It might be useful to note here that these data were recently qualified in SOCAT-v2020 (Bakker et al 2016, 2020). Why the data in 2008-2010 obtained by the same group and also available in SOCAT not included in this analysis ?

C-06: General: Why not adding other data available in SOCAT in this region to extend the analysis after 2012. In particular, a freshening has been observed in recent years in the subpolar zone (Holliday et al, 2020). This signal would decrease alkalinity and increase pCO2 (i.e. opposed to the effect of ALK discussed in the manuscript).

C-07: Page 6, Line 142: Takahashi et al 1993b: In reference Takahashi et al 1993a,b is listed twice.

C-08: Page 6, Line 155: The accuracy of 2 $\mu$atm is impressive for such analysis (ashore, correction to SST, etc...). In SOCAT the pCO2 data for 2006-2007 cruises have been assigned with a flag "C" (accuracy better than 5 $\mu$atm not 2 $\mu$atm).

C-09: Page 6, Line 158: For underway pCO2 cruises listed in Table S4 might be useful to specify the Expocodes (like in SOCAT, easier to see what cruises have been used).

C-10: Page 6, Line 159: Fig S1 ? (there is no figures in Supp Mat, this would help to see the way data in various water masses were selected).

C-11: Page 7, Line 169: Takahashi et al 1993b: In reference Takahashi et al 1993a,b is listed twice.

C-12: Page 7, Line 170: Authors use "pCO2" in the manuscript. In SOCAT, the data are in fCO2. For clarity, specify if you use pCO2 or fCO2 in DpCO2 calculations and figures (e.g. Figure 2).

C-13: Page 7, Line 171. I am not sure that Olafsson et al (2010) described the pCO2 underway observations in 2006-2007. Please clarify.

C-14: Page 7, Lines 173-175: Not sure to clearly identify when and where were conducted the cruises in 1983-2012 for the "Polar water collection". Are you using the discrete samples taken for pCO2 only or also use TALK and DIC data (and calculate pCO2) ? Maybe add a table in Supp Mat specifying date, location and what property was measured (pCO2, DIC, TALK). Are data obtained in 1983-1985 around 64N-28W included (Peng et al 1987) ?

C-15: Page 7, Line 175: "The data provide pCO2 for calculation of Delta pCO2 in Fig. 4." Data for figure 6 (not figure 4) ?

C-16: Page 7, Line 183: Equation number "1" used in previous section.

C-17: Page 8, Line 196: Maybe recall how Pw is calculated (e.g. Weiss and Price, 1980)

C-18: Page 8, Lines 1998-211: Not sure that all details on the way the wind were selected is useful. The most important results related to wind in this study is the gas transfer coefficient presented in Figure 4a (and should be also presented in figure 5 and 6, see comment below).

C-19: Page 9, Results: Figure 3 (if still published): add units on Y-axis (m2/s2). However, is figure 3 really useful here ? Maybe add this figure in Supp Mat. I think the "Results" section could start with the description of Figure 4.

C-20: Page 9, Line 240. Also discussed by Peng et al (1987).

C-21: Page 9, Line 240. Takahashi et al (1993). 1993a and b are the same in references.

C-22: Page 11: Table 1: Would be useful to add a column with climatological value in the same region (e.g. from Takahashi et al 2009). The climatology range is between 0 and -5 molC/m2/yr around Iceland, same range as listed in Table 1. Thus the regional difference depicted in Table 1 appears a permanent feature that supports the second part of the analysis (ALK).

C-23: Page 11, Lines 276-277: "possibly due to a weaker stratification (Fig 4b)". Figure 4b does not show change in stratification. Maybe add a reference here or show MLD variations.

C-24: Page 12, Line 280: typo: "fluxes"

C-25: Page 12 Line 281: "despite very different physical conditions". This is not specifically shown. Maybe recall the NAO shift and inform on observed SST, SSS, MLD variations in 1994-1995 ?

C-26: Page 12, Line 283: is the difference due to resolution of observation or real change between 1994-1995 and 2006-2007. Figure 5 shows much less seasonality in Arctic waters compared to Figure 4c. Is the difference of fluxes due to difference in pCO2 or wind or both ? Might be relevant here to present in Figure 5 (like for Figure 4) the gas transfer coefficient and DpCO2 (not only the fluxes).

C-27: Page 12: Line 296: Supp Fig S1: Fig S1 is not in the Supp Mat. Thus it is not easy to follow the selection of the data in Polar Water (see also comment C-10).

C-28: Page 12, Line 299: Why comparing the flux in Polar waters with the mean flux north of 50°N ? Might be more relevant to compare with the climatology from Takahashi et al (2009) in the same region, I.e. -3.5 to -4.5 molC/m2/yr around 68°N northeast off Iceland and seems coherent with the flux calculated with the pCO2 observations in 2006-2007 (see also comment C-22)

C-29: Page 12, Line 300: Authors write: "We evaluate the long term pCO2 characteristics of the three water masses from other data assembled over about 30 years". Which other data are used ? Is there a simple way to show which data are used in the 3 water masses ? Would be also interesting to use all SOCAT data available in this region to extend the analysis after 2012.

C-30: Page 12, Line 300: Typo "water masses"

C-31: Page 12, Line 304-306: Figure 6b also suggests that over 30 years, oceanic

pCO2 follows the atmospheric increase. A look at the monthly mean fCO2 in this region (from SOCAT-v2020) indicates an oceanic fCO2 trend of around 2.3 $\mu$atm/yr (here only for winter season and period 1982-2019).

C-32, Page 13, Figure 6a. Might be interesting to compare the DpCO2 seasonal cycle in "Polar water" with the climatology of Takahashi et al (2009) who probably used part of the same data to construct the climatology (?).

C-33, Page 13, Lines 320-322: Are you sure that Polar waters have higher TALK/DIC ratio (maybe a typo, higher DIC/TALK ?). Here, it would be interesting to show ALK/S and ALK/DIC ratios in various water masses (e.g. using your data and GLODAPv2 data)

C-34, Page 13, Lines 323-325: Lee et al (2006) did not discussed ALK in Arctic. Here, I would refer to Broullón et al (2019). Suggestion for a synthetic view of riverine ALK concentrations in Arctic: Fig S4 in Broullón et al (2019); they also conclude the difficulty to reconstruct ALK fields in this region.

C-35: Page 13, Line 325: When listing and discuss different TALK/S relationships it would be useful to show these relationships in a figure (and if possible colored by region). See also comment C-37. C-36, Page 14, Line 330. Maybe recall that Takahashi et al (2014) calculated the relation for PALK not ALK.

C-37: Page 14, Lines 332-333: Authors write: "However, the intercepts indicate considerable variability, they are higher than the average alkalinity of Arctic rivers and the intercepts are high in upstream regions of the East Greenland Current". Not easy to follow this statement. A plot of regional TALK/S relationship from both data and climatology would help.

C-38: Page 15, Line 350: ALK = 1048 $\mu$mol/kg. Recall this is a mean value (Cooper et al 2008). It might be relevant to associate an uncertainty to this value for the ALK flux estimates.

C-39: Page 15, Lines 353 and 359: Supplement ? there is no information in Supp Mat. (only Table S1-S4). Please check the file when resubmit the MS.

C-40: Pages 14-15, Lines 346-377: there are the main new results in this paper with calculation of ALK advected in the north Atlantic. I think authors could explain in more details their calculations (e.g. recall equation from Nondal et al 2009). A table with the ALK flux estimates (for the two calculations) and a schematic figure with boxes indicating Input (Sv), TALK concentrations and TALK budget would help to follow the discussion.

C-41: Page 15, Line 356: How the ratio DIC/TALK of 0.85 was chosen ? Based on the climatology (Takahashi et al 2014), the ratio DIC/TALK is always higher in this region (range 0.9-0.95). For a reader not familiar with this topic, why not showing a map of the DIC/TALK ratio in this region ?

C-42: Line 400: Authors write: "The Atlantic Water seasonal pCO2 variations we observe (Fig. 2c), are primarily driven by regional thermal and biological cycles". Probably Figure 4b not 2c. This is a well-known signal (e.g. Peng et al 1987; Takahashi et al 2002). Is it important to recall this in the conclusion ?

C-43: The conclusion is somehow very broad recalling pCO2 seasonality and anthropogenic CO2 in the North Atlantic (same as in the introduction, although anthropogenic CO2 is not discussed in the manuscript). In the conclusion authors should highlight the main findings and suggest what new observations or modeling experiment should be conducted to confirm their results or reduce the uncertainty in the calculations. For example are the Arctic and North Atlantic biogeochemical models include the effect of Arctic ALK changes to better simulate pCO2 and air-sea CO2 fluxes or is it a secondary process ?

C-44: Page 17: After "Authors contributions" add a section "Data availability" and list the links to the data used in this study (e.g. SOCAT, CARINA, GLODAP, other ?).

;;;;;;;;;; In references:

Line 577 and 580: Takahashi et al 1993 listed twice.

;;;;;;;;; Reference in this review not listed in the manuscript

Bakker, D. C. E., Pfeil, B., Landa, C. S., Metzl, N., O'Brien, K. M., Olsen, A., et al., 2016. A multi-decade record of high-quality fCO2 data in version 3 of the Surface Ocean CO2 Atlas (SOCAT), Earth Syst. Sci. Data, 8, 383-413, doi:10.5194/essd-8-383-2016.

Bakker, Dorothee C. E.; et al., (2020). Surface Ocean CO2 Atlas Database Version 2020 (SOCATv2020) (NCEI Accession 0210711). [indicate subset used]. NOAA National Centers for Environmental Information. Dataset. https://doi.org/10.25921/4xkx-ss49. Accessed [date].

Broullón, D., Pérez, F. F., Velo, A., Hoppema, M., Olsen, A., Takahashi, T., Key, R. M., Tanhua, T., González-Dávila, M., Jeansson, E., Kozyr, A., and van Heuven, S. M. A. C.: A global monthly climatology of total alkalinity: a neural network approach, Earth Syst. Sci. Data, 11, 1109–1127, https://doi.org/10.5194/essd-11-1109-2019, 2019.

Denvil-Sommer, A., Gehlen, M., Vrac, M. and Mejia, C., 2019. LSCE-FFNN-v1: a two-step neural network model for the reconstruction of surface ocean pCO2 over the global ocean, Geosci. Model Dev., 12, 2091-2105, https://doi.org/10.5194/gmd-12-2091-2019.

Fay, A. R. and McKinley, G. A.: Global open-ocean biomes: mean and temporal variability, Earth Syst. Sci. Data, 6, 273–284, doi:10.5194/essd-6-273-2014, 2014

Fröb, F., Olsen, A., Becker, M., Chafik, L., Johannessen, T., Reverdin, G., and Omar, A.: Wintertime fCO(2) Variability in the Subpolar North Atlantic Since 2004, Geophys Res Lett, 46, 1580-1590, 2019.

Holliday, N.P., Bersch, M., Berx, B. Chafik, L., Cunningham, S.,Florindo-Lopez, C., Hátún, H., Johns, W., Josey, S., A., Larsen, K., M., H., Mulet, S., Oltmanns, M.,

Reverdin, G., Rossby, T., Thierry, V., Valdimarsson, H.: Ocean circulation causes the largest freshening event for 120 years in eastern subpolar North Atlantic. Nat Commun 11, 585, doi:https://doi.org/10.1038/s41467-020-14474-y, 2020

Landschützer P., N. Gruber, and D. Bakker (2016), Decadal variations and trends of the global ocean carbon sink, Global Biogeochem. Cycles, 30, doi:10.1002/2015GB005359.

Landschützer, P., Laruelle, G. G., Roobaert, A., and Regnier, P.: A uniform pCO2 climatology combining open and coastal oceans, Earth Syst. Sci. Data Discuss., https://doi.org/10.5194/essd-2020-90, in review, 2020.

Metzl, N., A Corbière, G. Reverdin, A. Lenton, T. Takahashi, A. Olsen, T. Johannessen, D. Pierrot, R. Wanninkhof , S. R. Ólafsdóttir, J. Olafsson and M. Ramonet, 2010 Recent acceleration of the sea surface fCO2 growth rate in the North Atlantic subpolar gyre (1993 2008) revealed by winter observations, Global Biogeochem. Cycles, 24, GB4004, doi:10.1029/2009GB003658.

Peng, T.-H., Takahashi, T., Broecker, W.S., Olafsson, J., 1987. Seasonal variability of carbon dioxide, nutrients and oxygen in the northern North Atlantic surface water: observations and a model. Tellus-B 39, 5., 439–458

Rödenbeck, C., Bakker, D. C. E., Gruber, N., Iida, Y., Jacobson, A.R., Jones, S., Landschützer, P., Metzl, N., Nakaoka, S., Olsen, A., Park, G.-H., Peylin, P., Rodgers, K. B., Sasse, T. P., Schuster, U., Shutler, J. D., Valsala, V., Wanninkhof, R., Zeng, J., 2015. Data-based estimates of the ocean carbon sink variability – First results of the Surface Ocean pCO2 Mapping intercomparison (SOCOM). Biogeosciences 12: 7251-7278. doi:10.5194/bg-12-7251-2015

Sabine, C. L., Feely, R. A., Gruber, N., Key, R. M., Lee, K., Bullister, J. L., Wanninkhof, R., Wong, C. S., Wallace, D. W. R., Tilbrook, B., Millero, F. J., Peng, T.-H., Kozyr, A., Ono, T. and Rios, A. F.: The Oceanic Sink for Anthropogenic CO2, Science, 305(5682),

367–371, doi:10.1126/science.1097403, 2004

Terhaar, J., Kwiatkowski, L. & Bopp, L. Emergent constraint on Arctic Ocean acidification in the twenty-first century. Nature 582, 379–383 (2020). https://doi.org/10.1038/s41586-020-2360-3

Watson, A.J., Schuster, U., Shutler, J.D. et al. Revised estimates of ocean-atmosphere CO2 flux are consistent with ocean carbon inventory. Nat Commun 11, 4422 (2020). https://doi.org/10.1038/s41467-020-18203-3

Weiss, R. F. and Price, B. A.: Nitrous oxide solubility in water and seawater, Marine Chemistry, 8(4), 347–359, doi:10.1016/0304-4203(80)90024-9, 1980.

;;;;;;;;;;; End review

––––––––––––––––––––––––––––––––

---

## Referee Comment (RC2) · Anonymous Referee #2 · 23 Sep 2020

The paper presents an interesting summary of pCO2 data collected in different regions around Iceland (close Irminger Sea, Iceland Sea, and ice-free polar waters off or in the EGC), and in particular discusses the air-sea fluxes, in particular in winter. The ocean data are from different years, either from long time series stations or from seasonal surveys (with continuous pCO2 sampling) in 2006-2007. Then, theauthors discuss what could be the alkalinity properties and how water from the Arctic can cause a local large CO2 sink in the Iceland Sea.

I find the presentation of the pCO2 data sufficient and relevant. I would however object to the use of 'Irminger Sea' when discussing the results from the repeated station (IRM) southwest of Iceland. The station is located within the Irminger Current, in a region of often deep winter mixed layers, conditions that are far from common in the Irminger

[Figure]

Sea, even though other localized deep convection areas happen in its southwestern part, but only in specific years (see Fröb et al., 2018 and 2019), and involving different processes and water masses.

See also Reverdin et al. (2018) for a summary of conditions and trends in areas further southwest in the eastern Irminger Sea (rather close to Reykjanes ridge) and we show for some winters conditions favorable to a CO2-sink, albeit not for all, and with a tendency for a change from sink to source from the early to mi 1990s to the mid-2000s (the changes in the trends are more thurughly discussed in Leseurre et al., 2020)

I would also add some comments on interannual anomalies which are not really described, but certainly 2006-2007 are fairly remarkable years (see the red curve on Fig. 6, and also indications of anomalies in SST, temperature and probably winter mixed layers in the area southwest of Iceland).

My main concern with this paper is with the discussion on total alkalinity, which I dont find satisfactory. Definitely, the are complicated balances and processes happening in the Arctic and related to the exchanges with sea ice (either during its formation or later melt), and specific modes of primary production that take place iether in the ice, under the ice, or after its melt. Also, Arctic rivers can be high in TA, albeit by far the largest values are for the Canadian rivers, and this should not be such a large share of the fresh water flowing in the east Greenland Current, and furthermore entering the Iceland Sea. Thus, this component of the freshwater budget should not contribute to end members as high as 1700 micromol/kg. See for example the approach in Sutherland et al (2009) paper, and what is used in other more recent) papers combining alkalinity with water isotopes to investigate the freshwater budget both in the Arctic proper or in the East Greenland Current. Indeed the Nondal et al (2009) paper which find this resul is based on data from I.B. Oden in May 2002 (mostly within the dirfiting sea ice). This is a period of extensive (and thick) sea ice drifting from the Arctic, and thus the water is strongly influenced by the brine releases, and other Arctic processes (the interesting Rysgaard et al., 2007 study is not directly comparable to what is observed here). We

expect the sea ice to have a rather low alkalinity which would compensate the larger alkalinity (compared to salinity) of the brine enriched waters. The main issue is what part of the EGC water influences the surface Iceland Sea: is it the winter brine-enriched water or is it also the sea ice. Depending on that the result on the alkalinity properties of the surface water (independent on biological activity) is going to be very different.

Also, for the Iceland Sea, one would expect from Nondal et al (2009) that the 0-intersect is 582 micromol/kg for S > 34.5. This relationship is used for example in Fröb et al. (2019) for a nearby area in the Irminger Sea. This contrasts only slightly with Reverdin et al 2018 who find an intersect at 713 micromol/kg, but that's including data closer to Newfoundland and when there is no (or little) sea ice (the slightly higher values are coherent with the export from the Canadian Arctic, which freshwater component should be higher in TA). There is also plenty of new TA data in EGC for different seasons (not in winter) both in Nordic seas or Irminger Sea and east Greenland Current that could be looked at to get a better estimate of what Ta might be in inflowing polar water and in the Iceland Basin or nearby areas that could help on this issue (notice that the lower salinity there is not just resulting from input from the Arctic through exchange of freshwater and sea ice with the EGC, but also from excess precipitation in large parts of the Nordic Seas. The surface waters of the Icland Sea are also influenced by heat loss that will contribute to its undersaturation (this could be quantified, and discussed following different studies on the pCO2 budget in the Nordic Seas by the UIB group (A. Olsen, Bellery, Nondal...).

Finally, I had a hard time with the discussion of Fig. 7 and the two hypotheses formulated. It seems that only the horizontal arrow is discussed. I feel that the change of SSS and admixture with the EGC waters (both freshwater and melting sea ice) has to be associated with both changes in DIC and TA (not just TA). It is not fully clear to me how this would work out. To be convincing a simple box model should be established at least to provide an order of magnitude of what is proposed there, and whether this can explain the under-saturation, compared with other hypotheses.

Detailed comments: For Arctic water, results are interesting and show the all season strong undersaturation of these waters (in 2006-2007). Can maybe be related to studies in the Arctic proper? I would question a little bit the atmospheric value used here to estimate the atmospheric pCO2 in this area. More likely be higher atmos pressure than Reykjavik? but maybe CO2 in the northeasterlies closer to Greenland (if those are the conditions encountered) is a little lower than further east or south in Iceland Sea or Irminger Sea? (even if winds coming from Greenland...). Now I dont think that resulting differences would exceed 10 microatm... Also, it seems by reading the paper that the reference pCO2 (or fCO2) is not always the one measured in Iceland... There can be seasonally significant differences with other stations. How are these taken into account. Furthermore, I think that it is important to point that the measurements are only made in ice-free areas. This will seasonally vary, and not be always typical of the EGC (in particular in winter and spring). Either because these are situations when the sea ice could have melted (see above discussion of alkalinity) or on the other hand has been flushed away... At lest, this should be acknowledged, and taken into account in the discussion of the data.

l. 86: NAC waters derived from the Gulf Stream (I would add that they are highly transformed in the subpolar gyre by air-sea fluxes, but also admixture of subpolar gyre water; maybe less so at IRM end of the section)

l. 92: the mention of Arctic fresh water. Should add that only part of freshwater from Arctic (and in particular river input) brought back south by EGC/EGCC... (and in particular EGC...) l. 189: not c lear how the fluxes are estimated at this point although this is explained later. But there is a question on the error resulting of the use the monthly V**2 and instantaneous pCO2 measurements (which are linearly interpolated in time between successive cruises, if I understood correctly).

L. 193: replacing Westmann Islands values with Mauna Loa when CO2-ICE missing. Why Mauno Loa. What are the SLP values used then (Reykjavik values)? These changes can make big differences. WHen was CO2-ICE missing?

Arrows of plot 1 for currents: a bit schematic (in particular near Reykjanes Ridge and south of Iceland), but probably OK for the purpose

References Fröb, F., Olsen, A., Pérez, F. F., García-Ibáñez, M. I., Jeansson,E., Omar, A., and Lauvset, S. K.: Inorganic carbon and water masses in the Irminger Sea since 1991, Biogeosciences, 15, 51– 72, https://doi.org/10.5194/bg-15-51-2018, 2018.

Fröb, F., Olsen, A., Becker, M., Chafik, L., Johannessen, T., Reverdin, G. and Omar, A.: Wintertime fCO2 Variability in the Subpolar North Atlantic Since 2004, Geophys. Res. Lett., 46, 1580–1590, https://doi.org/10.1029/2018GL080554, 2019.

Reverdin, G., Metzl, N., Olafsdottir, S., Racapé, V., Takahashi, T., Benetti, M., Valdimarsson, H., Benoit-Cattin, A., Danielsen, M., Fin, J., Naamar, A., Pierrot, D., Sullivan, K., Bringas, F., and Goni, G.: SURATLANT: a 1993–2017 surface sampling in the central part of the North Atlantic subpolar gyre, Earth Syst. Sci. Data, 10, 1901– 1924, https://doi.org/10.5194/essd-10-1901-2018, 2018.

Sutherland, D. A., Pickart, R. S., Peter Jones, E., Azetsu‐Scott, K., Jane Eert, A., & Ólafsson, J. (2009). Freshwater composition of the waters off southeast Greenland and their link to the Arctic Ocean. Journal of Geophysical Research, 114, C05020. https://doi.org/10.1029/2008JC004808

---

## Author Comment (AC1) · 28 Oct 2020

Olafsdottir, Taro Takahashi, Magnus Danielsen and Thorarinn S. Arnarson

Author responses to Ref#1 comments We thank the referee for a thorough review and constructive suggestions for improvement. In our respone we first address general comments and follow those with specific comments.

General comments We respond to four issues raised by Ref#1 under General comments. 1 The referee introduces new and informative papers on the knowledge about the North Atlantic carbon chemistry and CO2 sink, the background of our study. This is

much appreciated and we take note of and incorporate these in the revised introduction section. 2 The referee expresses concern about the clarity of the data description. We address this, organize and expand section 2.1 Data acquisition. 3 The referee suggest we examine other data e.g. from SOCAT. The focus of our presentation is on DpCO2 which is evaluated from measurements of pCO2(sw) in discrete surface layer samples and from recorded underway pCO2(sw) measurements. We use no pCO2(sw) data calculated from other observed variables. The results give clear signs of the different sink/source characteristics of the three water masses examined. Inclusion of further data from e.g. SOCAT or GLODAP was therefore not considered advantageous for this presentation. This response relates also to specific comments C-06, C-29, C-31, C-33 and C-33. 4 Both Ref#1 and Ref#2 call for information on TCO2 and alkalinity to support the analysis. The discrete sea water sample pCO2 data we present generally include TCO2 results. ALK may thus be calculated. The call for information on TCO2 and ALK brings a complex temporal and biogeochemical variability in the Polar Water into focus. We intend to present another paper on that subject which we find too extensive to be added to the present paper. However, the referee suggestions may perhaps be met by adding summary information in figures of a kind similar to Fig. 6a and expand accordingly the Results and Discussion sections? That may still leave detailed observation materials for another presentation. This response relates also to specific comments C-14, C-33, C-35 and C-41.

Specific comments C-01 We agree with the referee that the carbonate chemistry processes in the Atlantic Water of the North Atlantic region and the CO2 sink are relatively well observed and understood compared to other ocean regions. Our aim is to present an unrecognized feature, the contribution of the Polar Water to the CO2 sink. The referee correctly notes that we have preferred to evaluate long term trends from winter observations when biological activity is low and a reference is added to the description of Fig.6b (Olafsson et al., 2009).

C-02 The referee brings our attention to new ways to define N-Atlantic regions or

biomes. We take note of these in the Introduction. We agree with the referee that comparison of seasonal data with recent pCO2 climatology might be suitable for another presentation.

C-03 The Holliday et al. (2020) description of freshwater in the N-Atlantic is certainly relevant and is added to the Introduction and the treatment on Fig. Fig.6b.

C-04 NAO describes the difference in the atmospheric pressure between the Icelandic Low and thge Azores High. Its relation with oceanographic variability in Iceland is weak (Ólafsson, 1999). The relations of NAO with the hydrographic conditions in the Iceland Sea 1994 – 1995 are discussed in two papers referred to (Flatau et al., 2003;Våge et al., 2015).

C-05, C-06 The 2006-2007 UW pCO2 covers one calendar year and was acquired with that as a goal and to cover the three main water masses. That effort was not extended for further years but UWpCO2 data assembled on later cruises in Icelandic waters will be used for further analysis and another presentation. We agree that hydrographic variations such as described by Holliday et al. 2020, are of importance. We take note of these in the description and discussion of the results in Fig. 6b. C-07, C-11 and C-21. Corrected

C-08 The accuracy of UW pCO2 measurements is evaluated in the SOCAT data processing. This is noted in 2.1 Data acquisition.

C-09 EXPOCODES have beed added to tables listing cruises.

C-10 Typo corrected by deleting: and S1. The UW data selection procedure is described on page 8, lines 212-217.

C-12 We never mention fCO2 in the manuscript. The consistent use of pCO2 is noted in secton 2.2 CO2 air-sea flux calculations.

C-13 Correct, lines 171 and 172 do not apply to underway measurements and have been deleted. A sentence on the UW pCO2 precision has been added.

C-14 The stations in the Polar Water collection are marked in Fig. 1. In Methods is a section on this data and another section is added the calculation of alkalinity. Data from 64N 28W, which is the IRM-TS location are not in this collection.

C-15 Corrected.

C-16 Corrected.

C-17 Reference added.

C-18 Lines 198 to 211 describe considerable work involved in processing data and flux calculations. Much more than how winds were selected. We prefer to keep these lines. A figure showing the gas transfer coefficients associated with the three water masses is added to Fig. 5.

C-19 Figure has been moved to Supp. Mat.

C 20 Reference added.

C-22 and C-28 A sentence comparing our results with the climatological Arctic and Atlantic Water fluxes is added. However, and as for C-02, comparison of seasonal data with recent pCO2 climatology might be suitable for another presentation.

C-23 and C-25 The intention here was to refer to Våge et al, 2015. Corrected. A figure showing the density variations with time and depth is added as a supplemenatry Figure S2.

C-24 Corrected

C-26 A figure with the gas transfer coefficient for the 2006-2007 UW periods has been added.

C-27 The Polar Water collection stations are shown in Fig. 1 and section has been added: 2.1.4 Polar Water data collection.

C-28 There is no previous assessment of the Polar Water annual flux. The comparison is to show how far the Polar Water flux is from the regional mean. A sentence comparing our results with the climatological Arctic and Atlantic Water fluxes is added.

C-29 The data section has been expanded to clearly describe the data sets used.

C-30 Corrected.

C-31 Agree and a sentence noting that the Atlantic Water pCO2 follows the atmospheric increase is added.

C-32 The relatively few Polar Water results in the climatology weigh little in the gridded presentation.

C-33 and C35 See General Comment 4. An overview covering the regional carbon chemistry variations is really beond the scope of this presentation.

C-34 Lee et al(2006) reference deleted and the important Broullón et al. (2019) added.

C-35 The Takahashi PALK presentation and its potential difference from ALK in the waters discussed is noted.

C-37 The sentence has been shortened to clarify the statement.

C-38 Cooper et al (2008) report flow weighted average alkalinity without uncertainty. Noted in the Discussion.

C-39 Not in Supplement but now added to text. See also C-40.

C-40 A table and equations explaining the estimates can be added,

C-41 The figure is intended as a basis for a discussion. It illustrates merely the effect of excess alkalinity on Atlantic Water, S:35 and t: 5°C, reaching the Nordic Seas. The figure gives no indication of how the excess is generated or how it acts in the transformation of Atlantic Water to Polar Water. We add elaborations on this issue in the Discussion.

C-42 Corrected figure number. In order to conclude on the state of the Atlantic Water

in comparison with Arctic and Polar Waters we expand the sentence: "The Atlantic Water seasonal pCO2 variations we observe are primarily driven by regional thermal and biological cycles but without much net influx of CO2".

C-43 We rewrite the Conclusions taking note of the constructive remarks.

C-44 Section on "Data availability" is added.

References Flatau, M. K., Talley, L., and Niiler, P. P.: The North Atlantic Oscillation, surface current velocities, and SST changes in the subpolar North Atlantic, Journal of Climate, 16, 2355-2369, 2003. Olafsson, J., Olafsdottir, S. R., Benoit-Cattin, A., Danielsen, M., Arnarson, T. S., and Takahashi, T.: Rate of Iceland Sea acidification from time series measurements, Biogeosciences, 6, 2661-2668, 2009.

Ólafsson, J.: Connections between oceanic conditions off N-Iceland, Lake Mávatn temperature, regional wind direction variability and the North Atlantic Oscillation, Rit Fiskideildar, 16, 41-57, 1999.

Våge, K., Moore, G. W. K., Jónsson, S., and Valdimarsson, H.: Water mass transformation in the Iceland Sea, Deep Sea Research Part I: Oceanographic Research Papers, 101, 98-109, http://dx.doi.org/10.1016/j.dsr.2015.04.001, 2015.

---

## Author Comment (AC2) · 28 Oct 2020

Olafsdottir, Taro Takahashi, Magnus Danielsen and Thorarinn S. Arnarson

Author response to Ref#2 comments We thank the referee for insightful comments on the oceanic regions studied and constructive suggestions for improvement.

General comments We respond to four issues raised by Ref#2 under General comments. 1 The referee objects to the use of the term Irminger Sea in connection with results from the time series station IRM. The oceanographic conditions in the Irminger Sea change when examined from south to north or from east to west and they also

change with time (Våge et al., 2011;Våge et al., 2009). We present data from a 5 station section 1993-1994 which includes IRM and from the IRM time series. In the Methods section. Line 134, we refer to this region as the "northern Irminger Sea" and consider it correct. The use of the IRM time series data is now clarified in the Methods section.

2 The referee points at $CO_2$ sink/source temporal variability in the Irminger Sea. We add a sentence on the long term variability to the presentation of Fig. 6b.

3 The referee is expresses concern with the discussion on total alkalinity. Both Ref#1 and Ref#2 call for information on TCO2 and alkalinity to support the analysis. The discrete sea water sample pCO2 data we present generally include TCO2 results. ALK may thus be calculated. The call for information on TCO2 and ALK brings a complex temporal and biogeochemical variability in the Polar Water into focus. We intend to present another paper on that subject which we find too extensive to be added to the present paper. However, the referee suggestions may perhaps be met by adding summary information in figures of a kind similar to Fig. 6a and expand accordingly the Results and Discussion sections? That may still leave detailed observation materials for another presentation.

4 The comments on Figure 7 are most relevant. The figure is intended as a basis for a discussion. It illustrates merely the effect of excess alkalinity on Atlantic Water, S:35 and t: 5°C, reaching the Nordic Seas. The figure gives no indication of how the excess is generated or how it acts in the transformation of Atlantic Water to Polar Water. We add elaborations on this issue in the Discussion.

Specific comments Line 86 Very relevant comment. A sentence and reference is added (Hátún et al., 2005) which includes IRM station observations.

Line 92 The words "a portion of" added.

Line 189 We use 30 day running means of U2 and interpolated atmospheric pCO2

numbers for calculating fluxes as outlined in the Wanninkhof papers referred to.

Line 193 The CO2-ICE data from the Vestmann Islands only extend back to 1992. Both CO2 ICE and Mauna Loa CO2 data are at 1 atm pressure. The CO2_ICE and Mauna Loa records were processed in an identical fashion. Periods where CO2-ICE is missing are added to Methods section.

Figure 1 We agree with the reviewer. Further details would not make the figure more informative.

References Hátún, H., Sandø, A. B., Drange, H., Hansen, B., and Valdimarsson, H.: Influence of the Atlantic Subpolar Gyre on the Thermohaline Circulation, Science, 309, 1841-1844, 2005.

Våge, K., Pickart, R. S., Thierry, V., Reverdin, G., Lee, C. M., Petrie, B., Agnew, T. A., Wong, A., and Ribergaard, M. H.: Surprising return of deep convection to the subpolar North Atlantic Ocean in winter 2007–2008, Nature Geoscience, 2, 67-72, 10.1038/ngeo382, 2009.

Våge, K., Pickart, R. S., Sarafanov, A., Knutsen, Ø., Mercier, H., Lherminier, P., van Aken, H. M., Meincke, J., Quadfasel, D., and Bacon, S.: The Irminger Gyre: Circulation, convection, and interannual variability, Deep Sea Research Part I: Oceanographic Research Papers, 58, 590-614, https://doi.org/10.1016/j.dsr.2011.03.001, 2011.

---

## Author Response (AR1)

Cover letter

Reykjavik 3rd December 2020.

Dear Peter Landschützer,

I thank you for very constructive comments on how to respond on the alkalinity issue and improve the manuscript. We agree with you on the need to provide further results and I hope you find the additional evidence on alkalinity we present sufficient. It is basically in Table 2, in Figure 6 and in the Results and Discussion sections.

The main text and the supplement have been extensively revised. The responses to reviewers 1 and 2 have been edited slightly to show how we have reacted to their comments. We submit the revised responses.

We look forward to hearing from you.

Yours sincerely,

Jón Ólafsson.

**Enhancement of the North Atlantic CO2 sink by Arctic Waters 1 2 Jon Olafsson1, Solveig R. Olafsdottir2, Taro Takahashi3,5, Magnus Danielsen2 and Thorarinn 3 S. Arnarson4,5 4 5 1 
[revised manuscript text omitted]

96

---

## Referee Report (RR1)

;;;;;;;;;;;;;

Review of the revised manuscript submitted to Biogeosciences, MS No.: bg-2020-313. "Enhancement of the North Atlantic CO2 sink by Arctic Waters" by  Olafsson et al.,

Review invitation: 2/1/21
Review accepted: 2/1/21
Review sent: 3/1/21

General comment:

Authors attempted to reply to almost all questions, suggestions from both reviewers. I find the revised version much easier to read, including the presentation of the data and methods, the results, discussion and interpretations. The conclusion has been also revised and the discussion indicates that these preliminary results certainly call for new analyses as mentioned by the authors in their first replies. Interesting new figures have been added (e.g. TCO2/S, TALK/S, new Figure 6) that should be redrawn (a curve for Atlantic water is apparently missing in these figures).

I think the paper is suitable for publication in its present form (pending few corrections).

I do have few comments (mainly minors) on the revised MS listed below.

;;;;;; Comments on revised manuscript

C-01: Line 25: In the abstract add uncertainty associated to the number listed (flux).

C-02: Line 58: "The North Atlantic is a relatively well observed region of the ocean". Lauvset et al (2016) reference is for a climatology. For pCO2, TALK and DIC surface observations in the North Atlantic maybe refer to Takahashi et al (2009), Bakker et al (2016) and Reverdin et al (2018) for recent observation synthesis in this region (see comment from reviewer 2).

C-03: Line 205: typo: carrbonic.

C-04: Line 301: typo: and and continued…

C-05: Figure 5a and 6: there is no data in January (correct ?).

C-06: Table 2, Figure 6 and text: for TCO2/S and TALK/S ratios specify units (e.g. µmol.kg-1/psu).

C-07: Figure 6: Red horizontal lines for Atlantic Waters specified in the legend but not seen in figures.

C-08: For a reader not familiar with this topic, would it be nice to show also TCO2 and TALK data (not only the ratio with S; maybe add plots of TCO2 and TALK data in sup mat).

C-09: Line 390: Reference to Figure 6e ?

C-10: Line 395: "Representative annual long term TCO2/S and ALK/S means would be more realistic but are not available". Would that be possible to get this information from GLODAP in this region or using TCO2 and TALK climatological fields (Takahashi et al 2014; Broullon et al 2019, 2020), although this might not change the results.

C-11: Line 399: reference to Table 1 or 2 ?

C-12: Line 402: Like in the submitted manuscript there is confusion here: "Polar waters having an increasingly higher alkalinity/salinity and alkalinity/TCO2 ratios". The ratio TCO2/ALK in Polar water is higher than in Atlantic (Table 2). Please rephrase.

C-13: Lines 410 and 448: Typo: Arcipelago

C-14: Line 434: Not clear to see in Figure S3 that pCO2 unchanged for the ratio 0.85. Could this be clarified ?

C-15: Line 445: Not sure how you derive the value of 34.8 10**12 mol CO2/yr and 0.058 PgC/yr. I suspect a typo here: maybe this is 4.8 10**12 mol CO2/yr (as correctly listed in the supplement).

C-16: Line 445: The effect of excess alkalinity of 0.058 PgC/yr is a number that could be recalled in the abstract. Would it be possible to associate an uncertainty for this number ?

C-17: Line 457: Typo: thTCO2/Alk

C-18: Figure S3: legend: typo: contours in in….

;;;;;;; end review

---

## Author Response (AR2)

**Responses to reviewers comments on the revised manuscript**

| Reviewer #1 comments | Response |
|---|---|
| C-01: Line 25: In the abstract add uncertainty associated to the number listed (flux). | Done |
| C-02: Line 58: "The North Atlantic is a relatively well observed region of the ocean". Lauvset et al (2016) reference is for a climatology. For pCO2, TALK and DIC surface observations in the North Atlantic maybe refer to Takahashi et al (2009), Bakker et al (2016) and Reverdin et al (2018) for recent observation synthesis in this region (see comment from reviewer 2). | Advice taken |
| C-03: Line 205: typo: carrbonic. | Corrected |
| C-04: Line 301: typo: and and continued… | Corrected |
| C-05: Figure 5a and 6: there is no data in January (correct ?). | Correct, yes |
| C-06: Table 2, Figure 6 and text: for TCO2/S and TALK/S ratios specify units (e.g. µmol.kg-1/psu). | Added µmol kg$^{-1}$ psu$^{-1}$ to Table 2 and to Figure 6 legends. |
| C-07: Figure 6: Red horizontal lines for Atlantic Waters specified in the legend but not seen in figures. | The red Atlantic Water benchmark lines are in a, b and c but the value 35.13 was above the scale in d).  Figure 6d  has been redraw with S scale 30-36 and a red benchmark line. |
| C-08: For a reader not familiar with this topic, would it be nice to show also TCO2 and TALK data (not only the ratio with S; maybe add plots of TCO2 and TALK data in sup mat). | These plots are intended for another ms on Polar Water. Can be added to sup. mat. if editor requests so. |
| C-09: Line 390: Reference to Figure 6e ? | Corrected |
| C-10: Line 395: "Representative annual long term TCO2/S and ALK/S means would be more realistic but are not available". Would that be possible to get this information from GLODAP in this region or using TCO2 and TALK climatological fields (Takahashi et al 2014; Broullon et al 2019, 2020), although this might not change the results. | GLODAP TCO2 and ALK data for Atlantic Water south of Iceland and north of 60°N is mostly for the summer months. Sentence left unchanged. |
| C-11: Line 399: reference to Table 1 or 2 ? | Table 2, corrected. |

| | |
|---|---|
| C-12: Line 402: Like in the submitted manuscript there is confusion here: "Polar waters having an increasingly higher alkalinity/salinity and alkalinity/TCO2 ratios". The ratio TCO2/ALK in Polar water is higher than in Atlantic (Table 2). Please rephrase. | Sentence simplified by removing Alk/TCO2 ratio. |
| C-13: Lines 410 and 448: Typo: Arcipelago | Both corrected |
| C-14: Line 434: Not clear to see in Figure S3 that pCO2 unchanged for the ratio 0.85. Could this be clarified ? | Figure S3 legend has been edited to clarify this. |
| C-15: Line 445: Not sure how you derive the value of 34.8 10**12 mol CO2/yr and 0.058 PgC/yr. I suspect a typo here: maybe this is 4.8 10**12 mol CO2/yr (as correctly listed in the supplement). | Typo corrected |
| C-16: Line 445: The effect of excess alkalinity of 0.058 PgC/yr is a number that could be recalled in the abstract. Would it be possible to associate an uncertainty for this number ? | Number has been added to the abstract. We argue that this number may be a minimum. There are too many poorly quantified variations (e.g. seasonal) and processes (e.g. ice formation and melting) for assessing uncertainty. |
| C-17: Line 457: Typo: thTCO2/Alk | Corrected |
| C-18: Figure S3: legend: typo: contours in in…. | Corrected |

| Reviewer #2 comments | Response |
|---|---|
| l. 49-51: end of sentence to rewrite | I am sorry, but cannot see what is suggested here. |
| l; 142: rewrite 'to 5 stations' as 'with 5 stations' | Done |
| l. 147: remove ', two years' | Done |
| l. 163: change 'so were also calibrated…( as 'the standards… measurements were similarly calibrated' | Done |
| l. 191: 'we use discrete samples of pCO2 and TCO2, and calculated…' | Done, sentence edited |
| l. 226: I would still have liked what is the percentage of missing data that had to be filled with Mauna Loa values. | The ICE records do not extend over all years of the long term observations. Mauna Loa data were used for 1983-1992 and |

| | |
|---|---|
| | 2010-2012. (Mace Head begins 1991). |
| l. 300: 'undersaturation' | Corrected |
| l. 338 'which confirms...' | Done |
| l. 465: 'western Arctic Ocean' (western is a little bit vague ... does it refer to the European Arctic?) | The wording is in the title of the Ouyang et al. reference. Clearly not European Arctic. |
| l. 477: carried | Corrected |
| l. 498: 'technical' | Corrected |
| Supplementary material: Figure S3, the caption title should include the mention that it is plotted at t=5°C and S=35 (then, it does not need to be mentioned in the next sentence for the red square) | Done. Figure S3 legend has been edited. Also in response to Ref #1 C-14 . |